# Expressive power of complex-valued restricted Boltzmann machines for solving non-stoquastic Hamiltonians

Chae-Yeun Park[1, 2] and Michael J. Kastoryano[1, 3, 4]

[1]*Institute for Theoretical Physics, University of Cologne, 50937 Köln, Germany*
[2]*Xanadu, Toronto, ON, M5G 2C8, Canada*
[3]*Amazon Quantum Solutions Lab, Seattle, Washington 98170, USA*
[4]*AWS Center for Quantum Computing, Pasadena, California 91125, USA*
(Dated: November 4, 2022)

Variational Monte Carlo with neural network quantum states has proven to be a promising avenue for evaluating the ground state energy of spin Hamiltonians. However, despite continuous efforts the performance of the method on frustrated Hamiltonians remains significantly worse than those on stoquastic Hamiltonians that are sign-free. We present a detailed and systematic study of restricted Boltzmann machine (RBM) based variational Monte Carlo for quantum spin chains, resolving how relevant stoquasticity is in this setting. We show that in most cases, when the Hamiltonian is phase connected with a stoquastic point, the complex RBM state can faithfully represent the ground state, and local quantities can be evaluated efficiently by sampling. On the other hand, we identify several new phases that are challenging for the RBM Ansatz, including non-topological robust non-stoquastic phases as well as stoquastic phases where sampling is nevertheless inefficient. We further find that, in contrast to the common belief, an accurate neural network representation of ground states in non-stoquastic phases is hindered not only by the sign structure but also by their amplitudes.

## I. INTRODUCTION

Over the past decade, Machine Learning (ML) has allowed for huge improvement not only in traditional fields such as image detection [1] and natural language processing [2] but also in other disciplines e.g. defeating human level in playing games [3, 4] and predicting protein structures [5]. Inspired by such successes, ML has been also actively applied for solving quantum physics problems. Examples include detecting phase transitions [6, 7] and decoding quantum error correcting codes [8–10]. However, arguably the most active contribution of ML to physics has been in the field of classical variational algorithms for solving quantum many body systems so called variational quantum Monte Carlo (vQMC).

A seminal study by Carleo and Troyer showed that the complex-valued restricted Boltzmann machine (RBM) [11] solves the ground state of the transverse field Ising and the Heisenberg models to machine accuracy. Subsequent studies demonstrated that other neural network based *Ansätze* such as convolutional neural networks (CNNs) [12, 13], and models with the autoregressive property [14, 15] also provide highly accurate solutions when combined with proper optimization algorithms. Despite these successes, the methods still suffer from several difficulties in solving highly frustrated systems [12, 16, 17].

Why then are some Hamiltonians so difficult to solve? In the path integral Monte Carlo, it is well known that stoquastic Hamiltonians – those with real-valued and non-positive off-diagonal elements – are tractable [18]. One of the crucial properties of stoquastic Hamiltonians is that the ground state is positive up to a global phase. As the RBM Ansatz also seems to solve stoquastic Hamiltonians well [11], a complex sign structure of the ground

state can be why it is difficult to solve a highly frustrated Hamiltonian with strong non-stoquasticity. Several studies [13, 17, 19] further support this claim by showing that training a neural network for a complex sign structure is demanding.

However, *expressivity* of the Ansatz has been largely overlooked in vQMC. As well-known universal approximation theorems [20–22] imply that a neural network with a sufficient number of parameters can express any physical functions, one may consider that *expressivity* is not really an issue. The theorem, nevertheless, tells little about how many parameters are required to represent a given function, and there are indeed some functions that a certain neural network fails to represent with a polynomial (in the input size) number of parameters [23]. Under a reasonable complexity theoretical assumption (that the polynomial hierarchy does not collapse), Ref. [24] also has shown that there is a quantum state $\Psi_{\mathrm{GWD}}$ with a local parent Hamiltonian whose amplitudes in the computational basis $|\Psi_{\mathrm{GWD}}(x)|^2$ cannot be obtained in a polynomial time. Recent numerical studies [25, 26] further show example quantum states whose outcome probability distributions are difficult even for large neural networks.

Thus, we do not have a clear argument why non-stoquastic Hamiltonians are difficult for vQMC albeit its formulation seems be to irrelevant to the target Hamiltonian. One can contrast with the density matrix renormalization group (DMRG) [27] which was first developed as an extension of numerical renormalization group, but subsequent numerical and theoretical studies have revealed that entanglement is the underlying principle behind this method. This connection explains why the DMRG works exceptionally well for one-dimensional gapped system but has difficulty solving higher dimensional systems. In comparison, we still do not have such

a good theory for vQMC.

With this in mind, we unveil a connection between the vQMC with complex RBMs and stoquasticity of the Hamiltonian. We thus aim to guide future theoretical studies on the mathematical foundation of the vQMC as well as numerical studies for Hamiltonians resilient to vQMC. Our investigation is based on the classification of three typical failure mechanisms: (i) **Sampling**: The sampling method, such as local update Markov chain Monte Carlo (MCMC), fails to produce good samples from the state, or the observables in the optimization algorithm cannot be accurately constructed from a polynomial number of samples. (ii) **Convergence**: The energy gradient and other observables involved in the optimization (e.g. the Fisher information matrix) can be accurately and efficiently obtained for each optimization step, yet the optimization gets stuck in a local minimum or a saddle point. (iii) The **expressivity** of the Ansatz is insufficient: the Ansatz is far from the correct ground state even for the optimal parameter set, i.e. $\min_\theta || |\psi_\theta\rangle - |\Phi_{GS}\rangle ||$ is large where $|\psi_\theta\rangle$ is a quantum state that Ansatz describe for a given parameter set $\theta$.

Specific pairs of Hamiltonian and Ansätze sometimes rule out one or several of the failure mechanisms. For instance, when the Hamiltonian has a known exact neural-network representation of the ground state (e.g. cluster state, toric code [28] and other stabilizer states [29, 30]), we can discard *expressivity* [case (iii)] as a failure mechanism. On the other hand, models with the autoregressive property [14, 15] are free from the MCMC errors as they always produce unbiased samples [31]. The Hamiltonians we consider in this work will typically not have known ground state representations and we mainly use the RBM Ansatz, as it is the best studied neural network Ansatz class and the most reliable performance [32]. Hence all three failure mechanisms can occur.

From those classifications, we set out to understand what role stoquasticity plays in the success and failure of variational Monte-Carlo with the RBM Ansatz. By way of example, we show that non-stoquasticity can cause problems with *sampling* [case (i)], while phase transitions within a non-stoquastic parameter region may yield *expressivity* problems [case (iii)]. We dub such a phase that cannot be annealed into a stoquastic parameter region "deep non-stoquastic phase." Given that "deep non-stoquastic phases" can be gapped, our observation implies that the dimension or gap of the system is not related to the reliability of the method in any straightforward manner. Rather, as for quantum Monte Carlo [18], the stoquasticity of the Hamiltonian is the more essential feature. Although our claim of a "deep non-stoquastic phase" seems to disagree with Ref. [17] which posits that even a shallow neural network (with depth 2 or 3) can express the ground state of non-stoquastic Hamiltonians, we show later in the paper that this is due to different levels of desired accuracy. We demonstrate that the *expressivity* problem indeed appears when the desired accuracy is as high as that of stoquastic ones. In addition,

we find that the *expressivity* problem is not only due to the sign structure of quantum states but also their amplitudes, where the latter dominates for system sizes up to about thirty for the models under study.

To identify difficult phases for vQMC, we utilize local basis transformations beyond the well-known Marshall-Peierls sign rule [33] for the $J_1$-$J_2$ models. We discover several basis transformations that reduce stoquasticity for XYZ-type Hamiltonians based on Ref. [34]. Although such transformations mitigate the *sampling* problem for a Hamiltonian connected to stoquastic phases, we show that they are not effective for Hamiltonians in deep non-stoquastic phases.

We still emphasize that our main concern in the paper is why errors from vQMC for non-stoquastic models are significantly larger than those for stoquastic models [12, 16, 35, 36]. However, such relatively large errors from non-stoquastic models may still be enough to obtain the ground state properties depending on the problem at hand (when the gap is much bigger than the errors even for large enough $N$). We thus do not claim a difficulty of a particular Hamiltonian (such as the two-dimensional $J_1$-$J_2$ model) for the vQMC; rather we want to understand *when and why* some Hamiltonians are relatively more difficult than others.

The remainder of the paper is organized as follows. We introduce the complex RBM wavefunctions and our optimization methods in Sec. II. We next establish our main observations in Sec. III by studying how non-stoquasticity affects the RBM using the one-dimensional XXZ and the $J_1$-$J_2$ models, the properties of which are well known. We then confirm our observations using a specially devised Hamiltonian in Sec. IV. We then resolve discrepancies between our and previous studies in Sec. V and conclude with the final remark in Sec. VI.

## II. VARIATIONAL MONTE CARLO AND COMPLEX-VALUED RESTRICTED BOLTZMANN MACHINE

Variational quantum Monte Carlo (vQMC) [37] is a classical algorithm for finding the ground state of a quantum Hamiltonian utilizing a variational Ansatz state. For a system with $N$ spin-1/2 particles (or qubits), vQMC considers an Ansatz state $\psi_\theta(x)$ where $x$ is the vector in the computation basis. When one can sample from $p(x) \propto |\psi_\theta(x)|^2$ and efficiently calculate the ratio $\psi_\theta(x')/\psi_\theta(x)$, the expectation value of a sparse observable $A$ can be obtained as

$$
\begin{aligned}
\langle A \rangle &= \sum_{x,x'} A_{x',x} \psi_\theta(x')^* \psi_\theta(x) \\
&= \sum_x |\psi_\theta(x)|^2 \sum_{x'} \frac{\psi_\theta(x')}{\psi_\theta(x)} A_{x',x} \\
&\approx \left\langle \sum_{x'} \frac{\psi_\theta(x')}{\psi_\theta(x)} A_{x',x} \right\rangle_{x \sim |\psi_\theta(x)|^2}
\end{aligned}
\tag{1}
$$

where $A_{x',x} = \langle x'|A|x \rangle$ and $\langle f(x) \rangle_{x \sim p(x)}$ is the statistical average of a function $f(x)$ over samples $\{x\}$ from $p(x)$. Likewise, one can also estimate the gradient of an observable $\nabla_\theta \langle \psi_\theta | A | \psi_\theta \rangle$, which enables one to stochastically optimize $\psi_\theta(x)$ toward an eigenstate with the minimum eigenvalue.

Initial studies [37, 38] have shown that this method solves ground states of several Hamiltonians when used with a proper parameterized wavefunction $\psi_\theta(x)$. However, choosing such a parameterized wavefunction had relied on several heuristics, and a general Ansatz was missing until Carleo and Troyer introduced the complex-valued restricted Boltzmann machine (RBM) quantum state Ansatz class [11] inspired by the recent successes in machine learning. As other machine learning approaches, the expressive power of the complex-valued RBM can be adjusted by increasing the number of parameters, and this Ansatz with a reasonable number of parameters solves the ground states of the transverse field Ising and the Heisenberg models in machine accuracy [11].

For complex parameters $a_i, b_j$ and $W_{ij}$ where $i \in [1, \cdots, N]$ and $j \in [1, \cdots, M]$, an (unnormalized) RBM state is given by

$$\widetilde{\psi}_\theta(x) = \sum_{y \in \{-1,1\}^N} e^{\sum_{i,j} w_{ij} x_i y_j + a_i x_i + b_j y_j}$$
$$= e^{\sum_i a_i x_i} \prod_j 2\cosh(\chi_j) \qquad (2)$$

where $\theta = (a, b, w)$ is the collection of all parameters, $x = (x_1, x_2, \cdots, x_N)$ is a basis vector in the computational basis (typically the Pauli $Z$ basis), $y = (y_1, y_2, \cdots, y_M)$ labels the hidden units, and the 'activations' are given by $\chi_j = \sum_i w_{ij} x_i + b_j$. We also introduce the parameter $\alpha = M/N$ that controls the density of hidden units and parameterizes the expressivity of the model. In addition, we will write $\psi_\theta(x) = \widetilde{\psi}_\theta(x) / \sum_x |\widetilde{\psi}(x)|^2$ to denote the normalized wavefunction.

For a given Hamiltonian, the parameters of the Ansatz can be optimized using a variety of different methods, including the standard second-order vQMC algorithm known as Stochastic Reconfiguration (SR) [37, 39] or a modern variant of the first order methods [15, 40, 41] such as ADAM [42]. Throughout the paper, we use the SR as it is believed to be more stable and accurate for solving general Hamiltonians [43]. At each iteration step $n$, the SR method estimates the covariance matrix $S$, with entries $S_{i,j} = \langle O_i^* O_j \rangle - \langle O_i^* \rangle \langle O_j \rangle$, and the energy gradient $f = \langle E_{\mathrm{loc}}^* O_i \rangle - \langle E_{\mathrm{loc}}^* \rangle \langle O_i \rangle$ where $O_i(x) = \partial_{\theta_i} \log[\widetilde{\psi}_\theta(x)]$ and $\langle \cdot \rangle = \sum_{x \sim |\psi_\theta(x)|^2} (\cdot)$ is the average over samples (see Refs. [11, 37] and Appendix A for details). The parameter set is updated as $\theta_{n+1} = \theta_n - \eta_n S^{-1} f$. In practice, a shifted covariance matrix $S' = S + \lambda_n \mathbb{I}$ with a small real parameter $\lambda_n$ is used for numerical stability. In the SR optimization scheme with the complex RBM, expectation values are obtained by sampling from the distribution $|\psi_\theta(x)|^2$, typically by conventional Markov chain Monte Carlo (MCMC). In some cases, we use the running averages of $S$ and $f$ when it increases the stability (i.e. we use $f_n = (1 - \beta_1) f_{n-1} + \beta_1 f$, $S_n = (1 - \beta_2) S_{n-1} + \beta_2 S$ for suitable choices of $\beta_1$, $\beta_2$ and update $\theta$ using $\theta_{n+1} = \theta_n - \eta_n S_n^{-1} f_n$).

To assess whether the sampling method works well, we introduce the exact reconfiguration (ER) that evaluates $S_{i,j}$ and $f$ from $\psi_\theta(x)$ by calculating the exponential sums $\sum_x |\psi_\theta(x)|^2 (\cdot)$ exactly, where $x$ is all possible basis vectors in the computational basis (thus we sum over $2^N$ or $\binom{N}{N/2}$ configurations depending on the symmetry of the Hamiltonian).

Within this framework, we classify the difficulty of ground state simulation as follows: We solve the system using the ER with $N = 20$ and the SR with $N = 28$ or $32$ (depending on the symmetry of the Hamiltonian). When the Hamiltonian is free from any of the problems [(i) sampling, (ii) convergence, (iii) expressivity], the converged energies from both methods will be close to the true ground state. If we observe that the ER finds the ground state accurately in a reasonable number of epochs [44], but SR does not, we conclude that the problem has to do with sampling. When there is a local basis change that transforms a given Hamiltonian into a stoquastic form, we apply such a transformation to see whether the *sampling* problem persists.

If both SR and ER fail, we evaluate the following further diagnostic tests: (a) We compare ER results from several different randomized starting points, and (b) we run the ER through an annealing scheme from a phase that is known to succeed. When all runs of ER return the the same converged energy, we conclude that the problem must be related to expressivity of the Ansatz. Otherwise, we try the annealing scheme as an alternative optimization method. Instead of training a randomly initialized RBM, we start from the converged RBM within the same phase and change the parameters of the Hamiltonian slowly. If the annealing with the ER also fails, we conclude that the *expressivity* problem is robust. Finally, we support the classification results from the above procedure by a scaling analysis of the errors for different sizes of the system.

## III. PRELIMINARY EXAMPLES

Stoquastic Hamiltonians [18] – those for which all off-diagonal elements in a specific basis are real and non-positive – typically lend themselves to simulation by the path integral quantum Monte Carlo method. In the path integral Monte Carlo, one evaluates the partition function $Z = \mathrm{Tr}[e^{-\beta H}]$ using the expansion

$$\mathrm{Tr}[e^{-\beta H}] = \sum_{x_0} \langle x_0 | (e^{-\frac{\beta}{K} H})^K | x_0 \rangle \qquad (3)$$

$$\approx \sum_{x_0, \cdots, x_{K-1}, x_K = x_0} \prod_{i=0}^{K-1} \langle x_{i+1} | (1 - \frac{\beta}{K} H) | x_i \rangle \qquad (4)$$

which is valid for large $K$. As all elements of $1-(\beta/K)H$ are non-negative when $H$ is stoquastic, the sum can be estimated rather easily. Likewise, one can also estimate the expectation value of an observable $A$ from a similar expansion of $\text{Tr}[Ae^{-\beta H}]/Z$. However, a "sign problem" arises when the condition is not satisfied, leading to uncontrollable fluctuations of observable quantities as the system grows.

The relevance of stoquasticity for the vQMC is far less explored, despite the fact that this method and its variants were introduced to alleviate the sign problem [39]. Although it is true that the vQMC is free from summations over alternating signs, the method still show several difficulties in solving frustrated Hamiltonians with a complex sign structure as argued in Ref. [16]. Given that the complex-valued RBM can represent quantum states with complex sign structure such as $\prod_{i,j} e^{i\phi_{i,j}\sigma_i^z\sigma_j^z}|+\rangle^{\otimes N}$ for arbitrary $\{\phi_{i,j}\}$ [24, 43], it is important to understand what makes complex-valued RBM struggle to solve non-stoquastic Hamiltonian.

In this section, we investigate this question in detail using the one-dimensional Heisenberg XXZ and $J_1$-$J_2$ models the properties of which are well known. Our strategy is simple. For each Hamiltonian, we use the original Hamiltonian and one with the stoquastic local basis, and observe how the local basis transformation affects the expressivity, convergence, and sampling. Throughout the paper, we will assume periodic boundary conditions for ease of comparison with results from the exact diagonalization (ED).

## A. Heisenberg XXZ and $J_1$-$J_2$ models

The Heisenberg XXZ model is given by

$$H_{\text{XXZ}} = \sum_i \sigma_i^x\sigma_{i+1}^x + \sigma_i^y\sigma_{i+1}^y + \Delta\sigma_i^z\sigma_{i+1}^z, \quad (5)$$

where $\sigma_j^{x,y,z}$ denote the Pauli operators at site $j$, and $\Delta$ is a free (real) parameter of the model. As this model is solvable by the Bethe Ansatz, it is well known that the model exhibits phase transitions at $\Delta = -1$ (the first order) and $\Delta = 1$ (the Kosterlitz–Thouless transition). Furthermore, the system is gapped when $|\Delta| > 1$ and in the critical phase when $-1 < \Delta \leq 1$.

The Marshall sign rule (applying the Pauli-$Z$ gate on all even (or odd) sites) changes the Hamiltonian into a stoquastic form in the Pauli-$Z$ basis:

$$H'_{\text{XXZ}} = \sum_i -\sigma_i^x\sigma_{i+1}^x - \sigma_i^y\sigma_{i+1}^y + \Delta\sigma_i^z\sigma_{i+1}^z \quad (6)$$

The Hamiltonian is then stoquastic regardless of the value of $\Delta$. Using the RBM with $\alpha = 3$, we plot the result from the ER and SR with and without the sign rule in Fig. 1(a) and (b). For the SR, we sample from the distribution $|\psi_\theta(x)|^2$ using the MCMC. As the system obeys the $U(1)$ symmetry and the ground states are

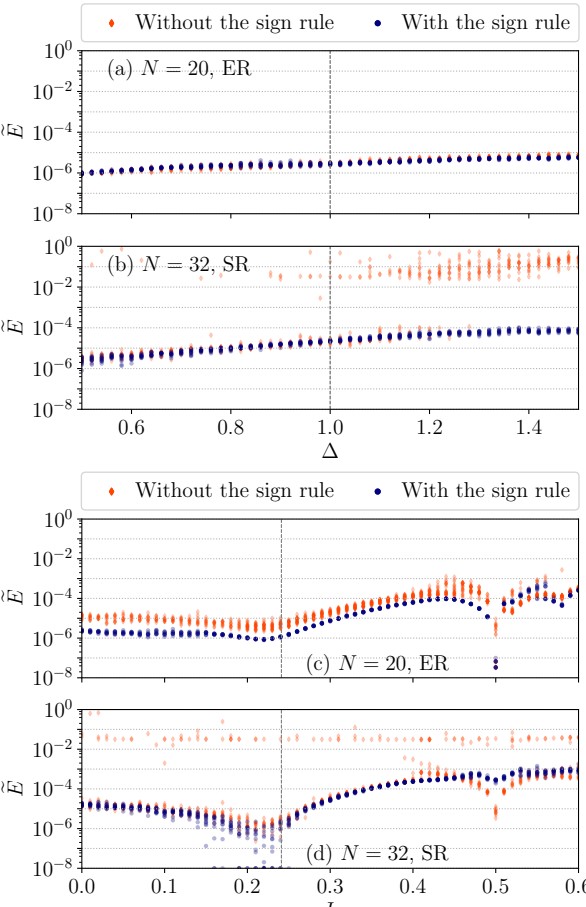

FIG. 1. Converged normalized energy $\widetilde{E} = (E_{\text{RBM}} - E_{\text{ED}})/E_{\text{ED}}$ as a function of model parameters for (a,b) the Heisenberg-XXZ and (c,d) $J_1$-$J_2$ model. For each model, the upper plots [(a) and (c)] present results for system size $N = 20$ with the Exact Reconfiguration method that optimizes the parameters using explicitly constructed wavefunctions from the RBM. The lower plots [(b) and (d)] show results from $N = 32$ with Stochastic Reconfiguration with Markov chain Monte Carlo sampling for optimization. The orange diamonds indicate simulation of the models in the original non-stoquastic basis, while the blue dots indicate simulations in the modified basis, after applying the Pauli-$Z$ operator on every even sites (the sign rule). Vertical dashed lines indicate where the KT-transitions take place ($\Delta = 1.0$ for the XXZ model and $J_2 \approx 0.2411$ for the $J_1$-$J_2$ model). For each value of the parameters, we have run the simulation 12 times and each point represents the result from a single run.

in the $J_z = \sum_i \sigma_i^z = 0$ subspace when $\Delta > -1$, we initialize the configuration $x$ to have the same number of up and down spins. For each Monte Carlo step, we update the configuration by exchanging $x_i$ and $x_j$ for randomly chosen $i$ and $j$. We further employ the parallel tempering method using 16 chains with different temperatures (see Appdendix A 2 for details) to reduce sampling noise. Likewise, we sum over the basis vectors in $J_z = 0$ for the ER. For each epoch, we use $|\theta| = NM + N + M$ number

of samples to estimate $S$ and $f$ unless otherwise stated.

Figure 1(a) clearly shows that the sign rule barely changes the results when we exactly compute the wavefunctions for optimization [45]. This can be attributed to the fact that the RBM Ansatz can incorporate the Pauli-$X, Y, Z$ gates as well as the phase shift gate $e^{-i\theta\sigma_k^z}$ for arbitrarily $\theta$ efficiently [46].

On the other hand, when we sample from the distribution [Fig. 1 (b)], some RBM instances fail to find the ground state without the sign rule, especially in the antiferromagnetic phase ($\Delta > 1.0$). Thus we see the *sampling* problem arises due to non-stoquasticity. Since the MCMC simply uses the ratio between two probability densities $|\psi_\theta(x')/\psi_\theta(x)|^2$, which is sign invariant, the *sampling* problem here has nothing to do with the ground state. Instead, it is caused by different learning paths taken by the original and the basis transformed Hamiltonians. When we use the original Hamiltonian $H_{\text{XXZ}}$, the learning ill-behaves when it hits a region of the parameter space $\theta$ where $S$ and $f$ are not accurately estimated from samples. The transformed Hamiltonian $H'_{\text{XXZ}}$ avoids this problem by following a different learning path [47]. We have observed that in general, the energy of a randomly initialized RBM is much closer to that of the ground state when the sign rule is applied and the learning converges in fewer epochs. We have further tested the SR without the sign rule using different sizes of the system $N = [20, 24, 28, 32]$ and up to $76,800$ samples for each epoch, but observed that the *sampling* problem persists regardless of such details. We also show that that this is not an ergodicity problem of the MCMC in Appendix B as the SR with the exact sampler (that samples from the probability distribution exactly constructed from $|\psi_\theta(x)|^2$) also gives the same results.

Next, let us consider the one-dimensional $J_1$-$J_2$ model, given by

$$H_{J_1-J_2} = \sum_i J_1\boldsymbol{\sigma}_i \cdot \boldsymbol{\sigma}_{i+1} + J_2\boldsymbol{\sigma}_i \cdot \boldsymbol{\sigma}_{i+2}. \qquad (7)$$

where we fix $J_1 = 1.0$. The Hamiltonian has a gapless unique ground state when $J_2 < J_2^*$ (thus, within the critical phase) and gapped two-fold degenerated ground states when $J_2 > J_2^*$. The KT-transition point is approximately known $J_2^* \approx 0.2411$ [48]. In addition, an exact solution at $J_2 = 0.5$ is known – the Majumdar-Ghosh point. The Marshall sign rule also can be applied to this Hamiltonian which yields:

$$H'_{J_1-J_2} = \sum_i J_1[-\sigma_i^x\sigma_{i+1}^x - \sigma_i^y\sigma_{i+1}^y + \sigma_i^z\sigma_{i+1}^z]$$
$$+ J_2\boldsymbol{\sigma}_i \cdot \boldsymbol{\sigma}_{i+2}. \qquad (8)$$

We note that this Hamiltonian is still non-stoquastic when $J_2 > 0$.

In Appendix C 1, we prove that on-site unitary gates that transform $H_{J_1-J_2}$ into a stoquastic form indeed do not exist when $J_2 > 0$. We also show that ground states in the gapped phase ($J_2 > J_2^*$) cannot be transformed into a positive form easily using the results from Ref. [49].

Simulation results for this Hamiltonian are presented in Fig. 1(c) and (d). First, as in the XXZ model, the ER results in Fig. 1(c) show that the sign rule is not crucial when we exactly compute the observables, i.e. the ER with and without the sign rule both converge to almost the same energies. However, in contrast to the XXZ model, there is a range of $J_2 \in (J_2^*, 0.5) \cup (0.5, 0.6)$ where all ER and SR instances perform badly (the error is $> 10^{-4}$ for some instances) even when the sign rule is applied [Fig. 1(c)]. It indicates that the *expressive power* of the network is insufficient for describing the ground state even though the system is gapped. We further show (see Appendix D) that this problem cannot be overcome by increasing the number of hidden units and revisit this issue in Sec. V using the supervised learning framework. Since this region cannot be annealed from a stoquastic point ($J_2 = 0$) without a phase transition, we argue that this parameter region is in a "deep non-stoquastic" phase.

When we use the SR, the results in Fig. 1(d) show that some of the instances always fail to converge to true ground states regardless of the Hamiltonian parameters when the sign rule is not applied. This is the behavior what we saw from the XXZ model that non-stoquasticity induces a *sampling* problem. On the other hand, when the sign rule is applied, all SR instances report small relative errors when $J_2 \leq J_2^*$ even though the transformed Hamiltonian is still non-stoquastic. We speculate that this is because the whole region is phase connected to the stoquastic $J_2 = 0$ point. We also note that previous studies [35, 36] using different variational Ansätze have reported a similar behavior of errors, suggesting that difficulty of "deep non-stoquastic" phases is not limited to the RBM Ansatz (see also Sec. V).

We summarize the results from the above two models with the following key observations.

**Observation 1.** *Complex RBMs represent ground states of spin chains faithfully when the Hamiltonian is stoquastic, up to a basis transformation consisting of local Pauli and phase-shift gates, or is phase connected to such a Hamiltonian.*

**Observation 2.** *There exists "a deep non-stoquastic phase", where the Hamiltonian cannot be locally or adiabatically transformed into a stoquastic Hamiltonian without crossing a phase transition. Complex RBMs have difficulty representing such ground states.*

**Observation 3.** *Sampling is stable along the learning path when the Hamiltonian is stoquastic or phase connected to a stoquastic Hamiltonian.*

In the next section, we will explore these observations more closely by introducing a more challenging example that combines all of the problems above.

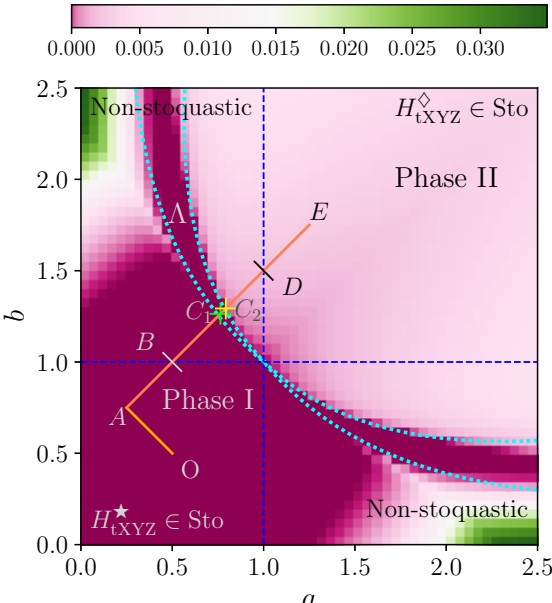

FIG. 2. Phase and stoquasticity diagrams of the twisted XYZ model. For parameters $0 \leq a, b \leq 2.5$, the difference between the lowest energies $(E_0 - E_1)/E_0$ in the symmetric and the anti-symmetric subspaces under the spin flip ($\sigma_z \leftrightarrow -\sigma_z$) for $N = 28$ is shown. As the ground states can break the $\mathbb{Z}_2$ symmetry in all three directions, we cannot determine phases solely from this plot. Thus we calculate magnetic susceptibilities in Fig. 3 and find that Phase I breaks the symmetry along the $z$-axis whereas Phase II recovers this. Between those two phases, Phase $\Lambda$ that breaks the symmetry along the $y$-axis appears when $a \neq b$. We depict approximate phase boundaries with dotted curves. On the other hand, dashed lines show stoquastic to non-stoquastic transitions. Local basis transformed Hamiltonians $H_{\text{tXYZ}}^{\diamond}$ and $H_{\text{tXYZ}}^{\star}$ are stoquastic in the first and third quadrants, respectively. In the second and forth quadrants, local (on-site) unitary gates that transforms the Hamiltonian into a stoquastic form do not exist. The untransformed Hamiltonian $H_{\text{tXYZ}}$ is stoquastic only when $a = b$ in this region. The line segment from $O = (0.5, 0.5)$ to $A = (0.25, 0.75)$ and $A$ to $E = (1.25, 1.75)$ indicate the parameters we simulate vQMC. The phase transitions along $\overline{AE}$ take place at $C_1 \approx (0.764, 1.264)$ and $C_2 \approx (0.793, 1.293)$.

## IV. FURTHER EXPERIMENTS

In previous examples, local basis transformations only marginally affected the expressive power of the model. Here, we introduce a Hamiltonian that involves the Hadamard transformation that cannot be embedded to the RBM Ansatz for a stoquastic transformation. The main findings in this section are (1) local basis transformations beyond the Pauli and phase-shift gates are useful for expressivity, (2) there is a conventional symmetry broken phase for which the RBM fails to represent the ground states, and (3) the number of samples to estimate observables correctly may scale poorly even for a

stoquastic Hamiltonian.

### A. Model Hamiltonian and phase diagram

We consider a next-nearest-neighbor interacting XYZ type Hamiltonian with "twisted" interactions:

$$H_{\text{tXYZ}} = J_1 \sum_{i=1}^{N} a\sigma_i^x \sigma_{i+1}^x + b\sigma_i^y \sigma_{i+1}^y + \sigma_i^z \sigma_{i+1}^z$$
$$+ J_2 \sum_{i=1}^{N} b\sigma_i^x \sigma_{i+2}^x + a\sigma_i^y \sigma_{i+2}^y + \sigma_i^z \sigma_{i+2}^z \quad (9)$$

where $a$ and $b$ are two real parameters. Note that $a$ ($b$) is the strength of $XX$ ($YY$) interaction for nearest-neighbors whereas it is on $YY$ ($XX$) interaction for next-nearest-neighbors. This particular Hamiltonian has a rich phase structure as well as stoquastic to non-stoquastic transitions. The stochastic regions do not coincide with the phases of the model. In addition, the system has $\mathbb{Z}_2$ symmetries in any axis $\sigma_i^{\{x,y,z\}} \leftrightarrow -\sigma_i^{\{x,y,z\}}$ as well as translational symmetry. Moreover, a $\pi/2$ rotation around the $z$-axis, i.e. $U = e^{-i\pi/4 \sum_i \sigma_i^z}$, swaps the parameters $a$ and $b$.

Throughout the section, we assume ferromagnetic interactions $J_1 = J_2 = -1$. As the Hamiltonian in this case becomes the classical ferromagnetic Ising model when $a = b = 0$, one may expect that the vQMC works well at least for small parameters. However, we will see that this intuition is generally misleading, as the non-stoquasticity of the model plays a very important role. We consider two other representations of the model, which are obtained by local basis transformations:

$$H_{\text{tXYZ}}^{\star} = J_1 \sum_{i=1}^{N} \sigma_i^x \sigma_{i+1}^x + b\sigma_i^y \sigma_{i+1}^y + a\sigma_i^z \sigma_{i+1}^z$$
$$+ J_2 \sum_{i=1}^{N} \sigma_i^x \sigma_{i+2}^x + a\sigma_i^y \sigma_{i+2}^y + b\sigma_i^z \sigma_{i+2}^z, \quad (10)$$

$$H_{\text{tXYZ}}^{\diamond} = J_1 \sum_{i=1}^{N} a\sigma_i^x \sigma_{i+1}^x + \sigma_i^y \sigma_{i+1}^y + b\sigma_i^z \sigma_{i+1}^z$$
$$+ J_2 \sum_{i=1}^{N} b\sigma_i^x \sigma_{i+2}^x + \sigma_i^y \sigma_{i+2}^y + a\sigma_i^z \sigma_{i+2}^z. \quad (11)$$

The Hamiltonian $H_{\text{tXYZ}}^{\star}$ ($H_{\text{tXYZ}}^{\diamond}$) is stoquastic for $0 \leq a, b \leq 1$ ($a, b \geq 1$), and can be obtained by applying $\pi/2$ rotation over $y$ ($x$) axes from the original Hamiltonian $H_{\text{tXYZ}}$, respectively. We note that as those rotations involve the Hadamard gate, they cannot be decomposed only by Pauli gates, e.g. $\pi/2$ rotation over the $y$-axis is given by $e^{-i\pi/4Y} = XH$. These Hamiltonians are obtained by applying the general local transformations

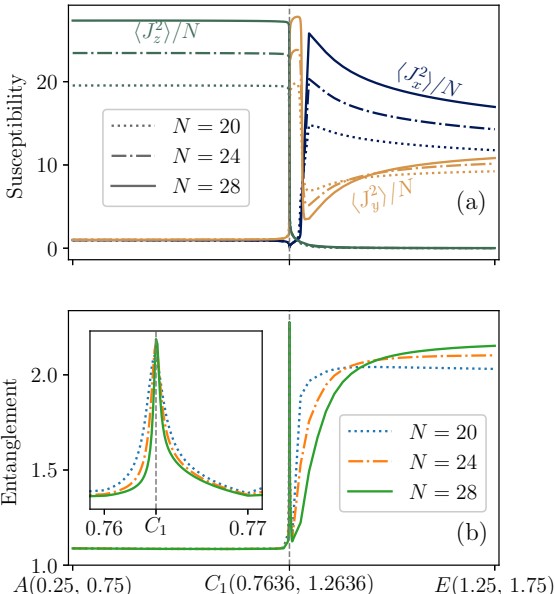

FIG. 3. (a) Magnetic susceptibility and (b) entanglement entropy for the ground state of the twisted XYZ model along the line $A = (0.25, 0.75)$ to $E = (1.25, 1.75)$. The result shows that there are three distinct phases. We locate the first phase transition point $C_1 \approx (0.7636, 1.636)$ that shows the divergence of entanglement entropy. The maximum values of the entanglement entropy are 2.16, 2.25, 2.28 for $N = 20$, 24, and 28, corroborating a logarithmic divergence of the entanglement entropy at criticality. In addition, we also observe polynomial decay of the correlation function $\langle \sigma_i^y \sigma_{i+k}^y \rangle$ in Phase II (see Appendix E for details).

described by Klassen and Terhal [34] to Eq. (9) (see Appendix C 2 for detailed steps). In addition, one may further transform $H_{\text{tXYZ}}^{\star}$ and $H_{\text{tXYZ}}^{\diamond}$ with local phase-shift gates $\prod_k e^{-i\pi/4\sigma_k^z}$ which can be embedded into the RBM Ansatz [46]. The resulting Hamiltonians are stoquastic for $a, b \geq 1$ and $0 \leq a, b \leq 1$, respectively, which are the reverse of ones before applying the phase-shift gates.

Before presenting vQMC results, let us briefly summarize the phase structure of the Hamiltonian that is presented in Fig. 2. To gain an insight, let us first consider the $a = b$ line. When $a = b < 1.0$, each term of the Hamiltonian prefers an alignment in $z$-direction so the ground state is $|\uparrow\rangle^{\otimes N} + |\downarrow\rangle^{\otimes N}$. Even though the $U(1)$ symmetry is broken when $a \neq b$, this ferromagnetic phase extends from $a = b < 1.0$ which we denote by Phase I in Fig. 2. On the other hand, the Hamiltonian prefers the total magnetization $J_z = \sum_i \sigma_i^z = 0$ when $a = b > 1.0$. The region of this phase is shown in Fig. 2 denoted by Phase II. As the total magnetization changes abruptly at $a = b = 1$ regardless of the system size, we expect a first order phase transition to take place at this point. However, the phase boundaries when $a \neq b$ are more complex and another phase $\Lambda$ appears in between two phases.

To characterize the phases when $a \neq b$, we plot magnetic susceptibilities and the entanglement entropy (von Neumann entropy of a subsystem after dividing the system into two equal-sized subsystems) along the line segment $\overline{AE}$ in Fig. 3. For each parameter $(a, b)$, we have obtained the ground state within the subspace preserving the $\mathbb{Z}_2$ symmetry along the $z$-axis using the ED (thus our ground states obey the $\mathbb{Z}_2$ symmetry even when the symmetry is broken in the thermodynamic limit). We see that the magnetic susceptibility along the $z$-axis diverges with the system size $N$ in Phase I which implies the symmetry will be broken when $N \to \infty$. Likewise, we also see that the symmetry along the $y$-axis is broken in Phase $\Lambda$. Furthermore, as entanglement entropy follows the logarithmic scaling at $C_1 \approx (0.764, 1.264)$ (see also Appendix E), we conclude that the phase transition at $C_1$ is the second order.

However, the signature of the phase transition from the entanglement entropy at $C_2$ is weak possibly because the phase transition is the infinite order Kosterlitz–Thouless transition. Thus we calculate the second derivative of the ground state energy in Appendix E and locate the second phase transition point $C_2 \approx (0.793, 1.293)$.

In addition, entanglement entropy shows that there is no hidden order in Phase I and $\Lambda$ as it is near to 1.0 which can be fully explained by the broken $\mathbb{Z}_2$ symmetry. We also see a signature of other phases when $a$ is small and $b$ is large (or vice versa), although we will overlook them as they are far from the parameter path we are interested in.

We note that even though the phases in Fig. 2 are identified following the conventional $\mathbb{Z}_2$ symmetry breaking theory, we will further restrict a symmetry class of the Hamiltonian when we discuss phase connectivity throughout the section as it provides a more consistent view. For example, we will consider that point $O$ (located on $a = b$ line which has the $U(1)$ symmetry) is not phase connected to $A$ (where the Hamiltonian obviously breaks the $U(1)$ symmetry), whereas $A$ and $B$ are phase connected. Our definition of phase connectivity is also compatible with a modern definition of phases in one-dimensional systems [50–52].

Finally, we depict the regions of stoquasticity in Fig 2. The model can be made stoquastic by a local basis change in the bottom left and top right quadrants. Within this phase diagram, we run our vQMC simulations along two paths $\overline{OA}$ and $\overline{AE}$. The path $\overline{OA}$ does not cross any phase or stoquasticity boundary, but it will show how a symmetry of the ground state affects the expressivity. On the other hand, the path $\overline{AE}$ crosses both the phase and stoquasticity boundaries thus will show how phase and stoquasticity transitions affect the vQMC.

## B. Variational Quantum Monte Carlo results

Our vQMC results for the twisted XYZ Hamiltonian are shown in Fig. 4. The shades in the middle of Fig. 4(c)

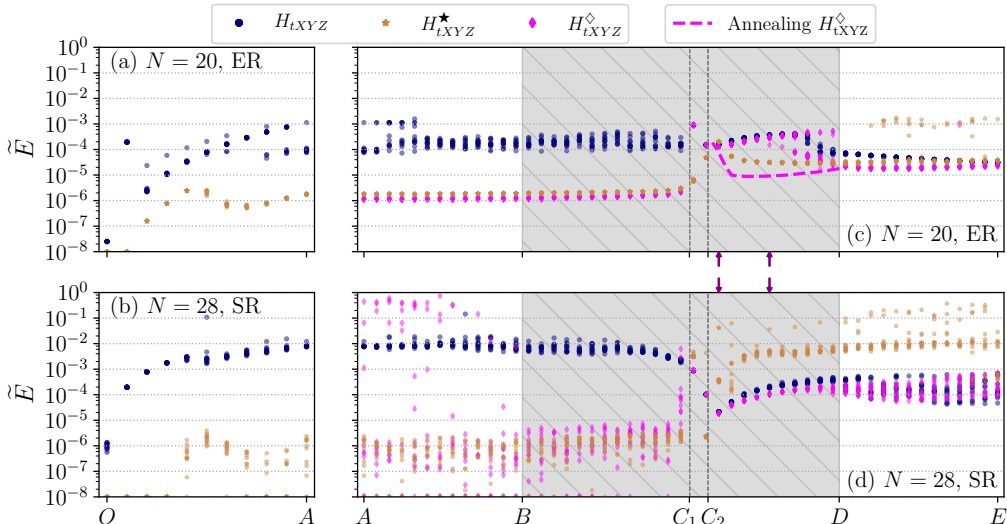

FIG. 4. Normalized energy from the vQMC for the twisted XYZ model. We have used the ER with $N = 20$ for (a) and (c), the SR with $N = 28$ for (b) and (d). (a,b) For $\overline{OA}$, the original Hamiltonian $H_{\text{tXYZ}}$ is only stoquastic at $O$ whereas $H_{\text{tXYZ}}^{\star}$ is stoquastic over the whole path. (c,d) The Hamiltonian $H_{\text{tXYZ}}$ is non-stoquastic over the whole path whereas $H_{\text{tXYZ}}^{\star}$ and $H_{\text{tXYZ}}^{\diamond}$ are stoquastic on the left and right side of the shaded region, respectively. In the shaded region, none of the Hamiltonians is stoquastic. Vertical dashed lines at $C_1$ and $C_2$ indicate the phase transition points. A dashed curve in (c) indicates an annealing result (see main text).

and (d) indicate the region where the model cannot be made stoquastic by a local rotation. We discuss the results for each path and phase below. As Phase $\Lambda$ (located between $C_1$ and $C_2$) is disconnected from others, we examine this case separately at the end of the section.

### 1. Path $\overline{OA}$ (Phase I)

As we have noted above, the ground state is a classical ferromagnet at location $O$. We observe that the RBM represents this state as expected. However, the error from the ER is getting larger as the parameter deviates from $O$. Since the Hamiltonian is always gapped along $\overline{OA}$, this result shows that non-stoquasticity affects vQMC even though the path does not close a gap; thus it reveals the importance of symmetry in the RBM expressivity.

With our symmetry sensitive definition of phase connectivity, the solubility indeed can be understood well as follows. First, the ground state at $O$ is represented by the RBM using the original Hamiltonian $H_{\text{tXYZ}}$ as it is stoquastic at this point (Observation 1). However, as going to $A$ breaks the $U(1)$ symmetry and it cannot be transformed into a stoquastic form only using local Pauli gates (see Appendix C 2), point $A$ is not guaranteed to be solvable using $H_{\text{tXYZ}}$. On the other hand, one can solve it using $H_{tXYZ}^{\star}$ which is stoquastic along the whole path $\overline{OA}$.

The fact that the transformed Hamiltonian $H_{tXYZ}^{\star}$ works much better than the original one $H_{\text{tXYZ}}$ in the

ER case contrasts to the XXZ and $J_1$-$J_2$ models where local Pauli rotations barely affected the expressive power. This is because the transformations in this model involve the Hadamard gate which is known to be challenging for the RBM [24, 46].

### 2. Path $\overline{AC_1}$ (Phase I)

We observe that both the transformed Hamiltonians $H_{\text{tXYZ}}^{\star}$ and $H_{\text{tXYZ}}^{\diamond}$ work better than the original Hamiltonian $H_{\text{tXYZ}}$ along the whole path when the ER is used [Fig. 4(c)]. Especially, the transformed Hamiltonians solve the ground state even for the shaded region ($\overline{BC_1}$) where none of the Hamiltonians are stoquastic. We explain this using the fact that $H_{\text{tXYZ}}^{\star}$ is stoquastic along $\overline{AB}$ and that there is no phase transition along $\overline{AC_1}$ (Observation 1). In addition, as applying local phase gates $\prod_k e^{-i\pi/4\sigma_k^z}$ which can be embedded into the RBM Ansatz transforms $H_{\text{tXYZ}}^{\diamond}$ into a stoquastic form (which is different from $H_{\text{tXYZ}}^{\star}$), Observation 1 also explains why $H_{\text{tXYZ}}^{\diamond}$ works. In contrast, the original Hamiltonian $H_{\text{tXYZ}}$ is non-stoquastic for all parameters in $\overline{AC_1}$.

On the other hand, the results from the SR [Fig. 4(d)] show that $H_{\text{tXYZ}}^{\star}$ which is stoquastic on the left side of the shaded region works better than $H_{\text{tXYZ}}^{\diamond}$. This result indicates that the MCMC is more stable when the stoquastic Hamiltonian is used, which is the behavior we have seen from the sign rule of the XXZ and the $J_1$-$J_2$ models (Observation 3). Interestingly, $H_{tXYZ}^{\diamond}$ appears to

be sensitive to the stoquastic transition although it is non-stoquastic throughout the path. We do not have a good explanation for this behavior.

### 3. Path $\overline{C_2 D}$ (Phase II)

We observe that $H^{\Diamond}_{\text{tYXZ}}$ which is stoquastic to the right of $D$ does not give the best result in this region $\overline{C_2 D}$ when the ER is used. However, a large fluctuation in the converged energies suggests that the *convergence* problem [case (ii)] arises, likely due to a complex optimization landscape. Thus we need to distinguish the problem between optimization and expressivity more carefully in this region.

For this purpose, we use an annealing approach as an alternative optimization method: We first take converged RBM weights for $(a, b) = (1.01, 1.51)$ (the point right next to $D$) where the Hamiltonian $H^{\Diamond}_{\text{tYXZ}}$ is stoquastic and run the ER from these weights instead of randomly initialized ones. We decrease the parameters $(a, b)$ of the Hamiltonian by $(0.01, 0.01)$ for each annealing step and run 200 ER epochs. The obtained results are indicated by a dotted curve in Fig. 4(c). The annealing result suggests that the expressivity is not the main problem up to the phase transition point $C_2$ (when considered from the right to the left).

The SR results show two noteworthy features compared to the ER results. First, the Hamiltonian $H^{\star}_{\text{tXYZ}}$ gives remarkably poor converged energies compared to the results from the ER. This result agrees with what we have seen from the sign rule: When the Hamiltonian is non-stoquastic, the learning path may enter a region where observables are not correctly estimated. Second, the shape of the curves from the Hamiltonians $H_{\text{tXYZ}}$ and $H^{\Diamond}_{\text{tXYZ}}$ are also different from that of the ER which is due to poor optimization. However, in Appendix F, we show that the *convergence* problem gets weaker as $N$ increases, thus the SR can solve the Hamiltonian $H^{\Diamond}_{\text{tXYZ}}$ in this region correctly for a large $N$ by examining the two parameter points of the Hamiltonian (indicated by arrows in Fig. 4).

We encapsulate the results in this region as follows: The vQMC works for $H^{\Diamond}_{\text{tYXZ}}$ that is phase connected to a stoquastic Hamiltonian even though it suffers from a optimization problem for small $N$. This result is consistent with Observation 1 and Observation 3.

### 4. Path $\overline{DE}$ (Phase II)

The ER results show that the RBM can express the ground state of all three Hamiltonians ($H_{\text{tXYZ}}$, $H^{\star}_{\text{tXYZ}}$, and $H^{\Diamond}_{\text{tYXZ}}$) wherein the stoquastic one in this parameter region $H^{\Diamond}_{\text{tYXZ}}$ works the best. One can also explain why some instances of the ER find the ground state of $H^{\star}_{\text{tXYZ}}$ using the existence of phase-shift gates ($\prod_k e^{-i\pi/4\sigma^z_k}$)

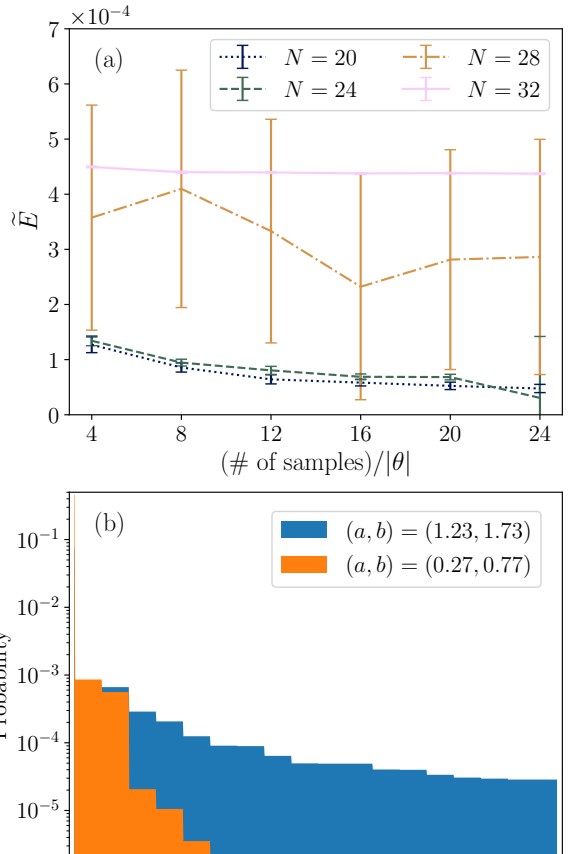

FIG. 5. (a) Normalized energy from the vQMC after convergence as a function of number of samples for different system sizes $N$. The Hamiltonian $H^{\Diamond}_{\text{tXYZ}}$ with $(a, b) = (1.23, 1.73)$ is simulated. Values in the $x$-axis indicate the number of samples used to estimate observables in each update step of the SR divided by $|\theta|$. The result is averaged over 12 vQMC instances and error bars indicate the standard deviation. Error bars for $N = 32$ are invisible as they are less than $2.0 \times 10^{-6}$. The result shows there is a transition in scaling near $N = 28$. (b) The first $10^3$ elements of $|\psi_{\text{GS}}(x)|^2$ where $\psi_{\text{GS}}(x)$ is the ground state of $H^{\Diamond}_{\text{tXYZ}}$ obtained from the ED when $N = 28$. Parameters $(a, b) = (0.27, 0.77)$ in Phase I and $(1.23, 1.73)$ in Phase II are used. When $(a, b) = (0.27, 0.77)$, the peak at the beginning indicate two largest elements of the distribution which are $\approx 0.458$. We see that the tail distribution for Phase II is much thicker. Moreover, the summation of the first $10^3$ elements is $\approx 0.998$ when $(a, b) = (0.27, 0.77)$ whereas it is only $\approx 0.294$ when $(a, b) = (1.23, 1.73)$. It suggest that one needs a huge number of samples to correctly estimate the probability distribution in Phase II.

that transforms the Hamiltonian into a stoquastic form. However, we do not have a nice explanation why non-transformed Hamiltonian $H_{\text{tXYZ}}$ also works in this region despite its non-stoquasticity.

On the other hand, the SR results show that the converged energies from $H_{\text{tXYZ}}$ and $H^{\Diamond}_{\text{tXYZ}}$ suffer large

fluctuations, which suggests that the *sampling* problem emerges. This result is unexpected as the Hamiltonian $H_{\text{tXYZ}}^{\diamond}$ is stoquastic in this region.

As the Hamiltonian is stoquastic, one may expect that using more samples easily overcomes the problem. However, this is not the case. To see this, we plot vQMC errors as a function of the number of samples in Fig. 5(a). Here, we have used $x \times |\theta|$ samples per epoch for each value in the $x$-axis. For $N = 20$ and 24, one observes that the errors get smaller as the number of samples increase. However, the results are subject to large fluctuations for $N = 28$, and gets worse when $N = 32$. Since $\theta$ itself scales as $\sim \alpha N^2$, our results show that this *sampling* problem is robust.

It is insightful to compare the *sampling* problem in this model to that in non-stoquastic models (such as the XXZ model in the antiferromagnetic phase without the sign rule). Even though they are both caused by a finite number of samples, the converged energies suggest that they have different origins. In the XXZ model, the converged normalized energies are mostly clustered above $10^{-2}$ regardless of the size of the system. On the other hand, they are below $10^{-3}$ and show clear system size dependency in this model. In Appendix G, we show that the *sampling* problem in this model only appears locally near the minima whereas it pops up in the middle of optimizations and spoils the whole learning procedure in non-stoquastic models.

The *sampling* problem occurring here is quite similar to the problem observed from quantum chemical Hamiltonians [53]. Precisely, Ref. [53] showed that optimizing the RBM below the Hartree-Fock energy for quantum chemical Hamiltonians requires a correct estimation of the tail distribution. However, the tail distribution of the ground state is often thick and a large number of samples are required to find the true optima. We find that the *sampling* problem of our model is also caused by such a heavy tail in the distribution. We can see this from the probability distribution of the ground states $|\psi_{\text{GS}}(x)|^2$ for $H_{\text{tXYZ}}^{\diamond}$ using two different parameters $(a, b) = (0.27, 0.77)$ and $(1.23, 1.73)$ which are deep in Phase I and II, respectively. We plot the first $10^3$ largest elements of $|\psi_{\text{GS}}(x)|^2$ for each parameter of the Hamiltonian in Fig. 5(b). The Figure directly illustrates that the distribution of the ground state in Phase II is much broader than that of Phase I. Moreover, the sum of the first $10^3$ elements is only $\approx 0.294$ in Phase II, which implies that one needs a huge number of samples to correctly estimate the probability distribution.

### 5. Phase Λ

Finally, we show that the RBM has difficulty representing the ground states in phase Λ even though it is a simple conventionally ordered phase (which was observed from the entanglement entropy). We simulate vQMC along the line $\overline{JK}$ in Fig. 6(a) which is deep in phase Λ. The

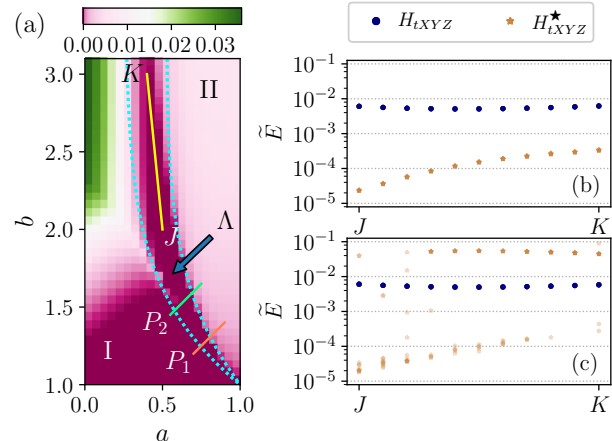

FIG. 6.   (a) The second quadrant of the phase diagram in Fig. 2. We calculate the second derivative of the ground state energy along paths $P_1$ and $P_2$ in Appendix E to locate the phase transition points. Converged normalized energy using (b) the ER with $N = 20$ and (c) the SR with $N = 28$ along the path $\overline{JK}$ where $J = (0.5, 2.0)$ and $K = (0.4, 3.0)$.

transformed Hamiltonian $H_{\text{tXYZ}}^{\diamond}$ gives almost the same converged energies as the original Hamiltonian $H_{\text{tXYZ}}$, so we do not present them in Fig. 6(b) and (c). The ER results in Fig. 6(b) clearly show that the error increases as we go deeper in this phase. As in the $J_1$-$J_2$ model, we simulate the ER with varying $N$ and different numbers of hidden units at point $K$ in Appendix H which confirms that there is the *expressivity* problem in this phase. In addition, the SR results [Fig. 6(c)] show that the *sampling* problem also arises when we use $H_{\text{tXYZ}}^{\star}$ which performed best with the ER. We also note that we cannot apply the results in Ref. [49] as in the $J_1$-$J_2$ model, since there is no hidden orders in this phase.

To summarize overall the results from the twisted XYZ model, we have found that Observation 1 and 2 hold in general by examining the behavior of the vQMC in different phases and stoquastic/non-stoquastic regions. In addition, we also have observed that a different type of *sampling* problem may arise even when solving a stoquastic Hamiltonian. Thus we modify our observation 3 slightly as follows:

**Observation 3** (Second version). *Sampling is stable along the learning path when the Hamiltonian is stoquastic or phase connected to a stoquastic Hamiltonian. However, the number of samples required to converge may scale poorly even when the Hamiltonian is stoquastic.*

## V. EXPRESSIVITY PROBLEM FROM SUPERVISED LEARNING

We point out that observation 2 conflicts with the assertion [17] that a shallow neural network (with depth 2 or 3) can express the ground state of a frustrated system

| Layer | Kernel size | Size of feature map |
|---|---|---|
| Input | - | $N \times 1$ |
| Conv-1 | $\lfloor N/2 \rfloor + 1$ | $N \times W/2$ |
| Even activation | - | - |
| Conv-2 | $\lfloor N/2 \rfloor + 1$ | $N \times W$ |
| Odd activation | - | - |
| Mean | - | $1 \times W$ |
| Fully connected | - | 1 |

TABLE I. Architecture of neural networks used for the supervised learning. We use two convolutional (Conv-1,2) and one fully connected layers without biases. A periodic padding is used for convolutional layers, so they commute with the translation of the input. The kernel size $\lfloor N/2 \rfloor + 1$ is used both for Conv-1 and Conv-2. Numbers of input/output channels are $(1, W/2)$, $(W/2, W)$ for Conv-1 and Conv-2, respectively, where $W$ is a parameter that determines the width of the network. We use an even (odd) activation function after Conv-1 (Conv-2), to preserve the $\mathbb{Z}_2$ symmetry. See Appendix I for further details.

without problem. Precisely, the authors have trained a neural network to reproduce amplitudes and signs of the ground state obtained from the ED without imposing the sign rule and found that the reconstructed states give a high overlap with the true ground state; the statement "expressibility of the Ansätze is not an issue—we could achieve overlaps above 0.94 for all values of $J_2/J_1$" is given.

However, we consider a 0.94 overlap to be insufficient evidence for this claim. For example, we have obtained an overlap between of $\gtrsim 0.999$ for the one dimensional $J_1$-$J_2$ model when $J_2 = 0.44$ and $N = 20$ with a non-stoquastic ansatz, but an order of magnitude better with a stochastic one.

In this section, we further clarify the discrepancy by showing that the expressivity problem appears even in the supervised learning set-up (as in Ref. [17]) that is less prone to other problems (sampling and training) when the desired accuracy gets higher. Notably, we show that, in contrast to what one might expect, a neural network (even after taking the symmetries into account) has a problem in reproducing *amplitudes* of a deep non-stoquastic ground state.

### A. Neural networks and learning algorithms

We use a convolutional neural network (Table I) for the supervised learning experiment. Our network is invariant under translation, as the convolutional layers commute with translation and outputs are averaged over the lattice sites before being fed into the fully connected layer. We further embed the $\mathbb{Z}_2$ symmetry in the network by turning off all biases and using even and odd activation functions after the first and second layers, respectively, following Refs. [19, 54]. We introduce a parameter $W$ that characterizes the width of the network. We further use $\theta$ to denote a vector of all parameters and $f_\theta(x)$ for

the output of the network. We note that our network structure is very close to a convolutional network used in Ref. [17].

We utilize this network to learn the amplitudes and signs of the true ground states obtained from the original Hamiltonian $H_{J_1-J_2}$ (without imposing the sign rule). We use the kernel size $= \lfloor N/2 \rfloor + 1$ for the convolutional layers as smaller kernels have failed to reproduce the sign rule for $J_2 = 0$ (when the sign rule is correct for all configurations). The number of parameters of the network is then given by $(W/2 + W^2/2)(\lfloor N/2 \rfloor + 1) + W$. For $N = 24$, the networks with $W = 16$ and $32$ have $1,784$ and $6,896$ parameters respectively.

Our learning set-up is slightly different from that of Ref. [17]: (i) Instead of mimicking the amplitudes, we use our network as an energy based model to be sampled from and (ii) we train our network using the whole data (all possible configurations $x$) without dividing the training and validation sets, as we are only interested in expressivity, not generalization property of the network.

#### 1. Learning amplitudes

To model the amplitudes $|\psi_{\mathrm{GS}}(x)|^2$, we use the output of our network $f_\theta(x)$ as the energy function for the energy based model. Even though there are models with the autoregressive property, which are easier to train, we choose the energy based model as it does not add any additional constraints (such as ordering of sites) and symmetries can be naturally imposed. Precisely, we model the amplitudes with $p_\theta(x) = e^{f_\theta(x)}/Z$ where $Z = \sum_x e^{f_\theta(x)}$ is the partition function of the model. We note that even though $Z$ is intractable in general, we can compute $Z$ for system sizes up to $N = 28$ rather easily thanks to the symmetry imposed on the network. Our loss function is the cross entropy $l(\theta) = -\sum_x p_{\mathrm{data}}(x) \log[e^{f_\theta(x)}/Z]$ where the probability distribution for data we want to model captures the amplitudes of the ground state: $p_{\mathrm{data}}(x) = |\psi_{\mathrm{GS}}(x)|^2$.

The gradient of the loss function can be estimated using samples from the data and model distributions as

$$g \approx -\{\mathbb{E}_{p_{\mathrm{data}}(x)}[\nabla_\theta f_\theta(x)] - \mathbb{E}_{p_\theta(x)}[\nabla_\theta f_\theta(x)]\}. \quad (12)$$

For the energy based model, estimating the second term is difficult in general as we need to sample from the model using e.g. MCMC. However, we here sample exactly from $p_\theta$ which is again possible up to $N = 28$. We use the same number of samples (the mini-batch size) from $p_{\mathrm{data}}(x)$ and $p_\theta(x)$ to estimate the first and the second terms.

Unfortunately, we have found that usual first-order optimization algorithms such as Adam [42] do not give a proper minima due to a singularity of the ground state distribution [43] (see also discussions in Ref. [55]). Thus we have utilized the natural gradient descent [56] (which can be regarded as a classical version of the SR) to optimize our energy based model, which is tractable up to

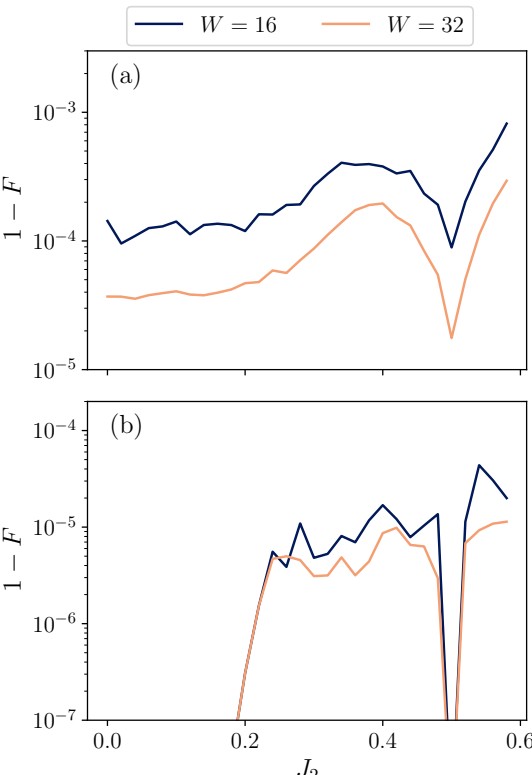

FIG. 7. Infidelity $1-F = 1-\langle\psi_{\mathrm{GS}}|\psi_{\mathrm{recon}}\rangle^2$ between the true ground states and reconstructed states as a function of $J_2$ for the one-dimensional $J_1$-$J_2$ model is shown. We train neural networks to reproduce (a) the amplitudes and (b) the signs of the ground states. The systems size $N = 24$ and the widths of network $W = 16$ and $32$ are used. To train the amplitude network, the natural gradient descent with hyperparameters $\eta = 10^{-4}$, $\beta_1 = 0.9$, $\beta_2 = 0.999$ and the mini-batch size 1024 are used. For the sign network, we use Adam optimizer with the learning rate $\eta = 2.5 \times 10^{-5}$, the mini-batch size 32 (see Appendix I for details).

several thousands of parameters. Precisely, we compute the classical Fisher matrix $\mathcal{F} = (\mathcal{F}_{ij})$ where

$$
\begin{aligned}
\mathcal{F}_{ij} = {}& \mathbb{E}_{p_\theta(x)}[\partial_{\theta_i} f_\theta(x) \partial_{\theta_j} f_\theta(x)] \\
& - \mathbb{E}_{p_\theta(x)}[\partial_{\theta_i} f_\theta(x)] \mathbb{E}_{p_\theta(x)}[\partial_{\theta_j} f_\theta(x)]
\end{aligned} \tag{13}
$$

each epoch and update parameters $\theta$ as $\theta_{n+1} = \theta_n - \eta_n(\mathcal{F}_n + \lambda_n \mathbb{1})^{-1} g_n$. Here, $\eta_n$ and $\lambda_n$ are the learning rate and the (epoch dependent) regularization constant, respectively. We also use a momentum for the gradient $g$ and the Fisher matrix $\mathcal{F}$ to stabilize the learning procedure, i.e. $g_n = \beta_1 g_{n-1} + (1 - \beta_1)g$ and $\mathcal{F}_n = \beta_2 \mathcal{F}_{n-1} + (1 - \beta_2)\mathcal{F}$.

To quantify the expressivity of the network, we record the overlap between reconstructed and the true ground states (assuming that the sign is correct) over the training, which can be expressed as $\langle\psi_{\mathrm{GS}}|\psi_{\mathrm{recon}}\rangle = \sum_x |\psi_{\mathrm{GS}}(x)|e^{f_\theta(x)/2}/\sqrt{Z}$. As the network obeys the same symmetry as the ground state, this quantity can be also calculated only by summing over the symmetric configurations.

#### 2. Learning signs

We use the same network to model the sign structure. As the problem nicely fits into the binary classification problem, we feed the output of the network into the sigmoid function and use it to model the sign structure, i.e., we use $P[\psi_{\mathrm{GS}}(x) > 0] = \mathrm{Sigmoid}(f_\theta(x))$. We then optimize the network by minimizing the binary cross entropy

$$
\begin{aligned}
l(\theta) = {}& -\sum_x p_{\mathrm{data}}(x)\big\{y_{\mathrm{data}}(x)\log[\mathrm{Sigmoid}(f_\theta(x))] \\
& + (1 - y_{\mathrm{data}}(x))\log[1 - \mathrm{Sigmoid}(f_\theta(x))]\big\}
\end{aligned} \tag{14}
$$

where $y_{\mathrm{data}}(x) = 1$ when $\psi_{\mathrm{GS}}(x) > 0$ and $y_{\mathrm{data}}(x) = 0$ otherwise. In practice, we estimate the loss function and its gradient using samples from $p_{\mathrm{data}}(x)$.

We have found that usual first-order optimizers such as Adam properly find optima in this case. We also compute the overlap between the true ground state and the reconstructed quantum state $|\psi_{\mathrm{recon}}\rangle = \sum_x \mathrm{sgn}[f_\theta(x)]|\psi_{\mathrm{GS}}(x)|\,|x\rangle$, where $\mathrm{sgn}(x)$ is the sign function.

### B. Results

We show the converged infidelity $1-\langle\psi_{\mathrm{GS}}|\psi_{\mathrm{recon}}\rangle^2$ from neural networks trained for the amplitudes and signs in Fig. 7. The results are obtained after tuning hyperparameters and initializations the detail of which can be found in Appendix I. The results show that the infidelity from a "deep non-stoquastic" phase is larger than that of stoquastic case both for Fig. 7(a) and (b) where we have trained the amplitudes and signs, respectively. However, the errors go up to $\approx 1.95 \times 10^{-4}$ even for $W = 32$ (where the number of parameters is $6,896$) when we train the amplitudes ( assuming the signs are correct) whereas the typical errors are smaller than $10^{-5}$ when we do the opposite. Thus our result strongly suggests that learning amplitudes is more difficult for a neural network than learning the sign structure. As we sampled from the distribution exactly and we expect that the learning procedure is more reliable in the supervised learning set-up, we conclude that this is due to lack of *expressivity* [case (iii)] of a neural network. We also note that solving the linear equation $(\mathcal{F}_n + \lambda_n)v = g_n$, which requires $O(|\theta|^{2-3})$ operations, dominates the learning cost. This is the main limiting factor that prevents us from using a bigger network.

We further plot scaling of errors for different sizes of the system using $J_2 = 0.0$ and $0.4$ in Fig. 8(a). We see that both errors from the amplitude and the sign networks increase with $N$ regardless of $J_2$. We also see that

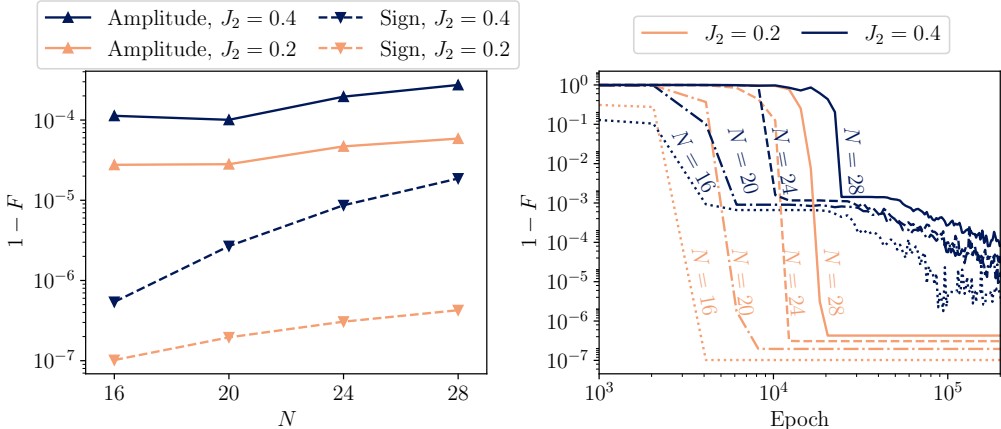

FIG. 8. For neural networks with $W = 32$ and chosen hyperparameters (see Appendix I for details), we plot (a) scaling of the converged infidelities for different sizes of system $N = [16, 20, 24, 28]$ and (b) initial learning curves (results from the first $2 \times 10^5$ epochs whereas we have trained the network in total $\approx 4.10 \times 10^6$ epochs) from the sign network with different $N$.

the errors from the amplitude network are significant up to the system size $N = 28$, but the slope of the sign network is slightly sharper. We still leave a detailed scaling behavior for future work as our simulation results here are limited up to the system size $N = 28$.

We also show initial learning curves from the sign network in Fig. 8(b). Consistent with a previous observation [54], learning the sign structure takes some initial warm-up time when the sign rule is not used. We conjecture that this is also related to the generalization property observed in Ref. [17]. In Appendix I 4, we further show that imposing the sign rule makes the initial warm-up time disappear but does not change the overall performance.

We note that, as we can use the first order optimization algorithms, increasing the size of network as well as training longer epochs are much easier for the sign networks than for the amplitude networks. We thus believe that increasing the system size while maintaining the accuracy in the supervised learning set-up is mostly obstructed by the amplitude networks (under the assumption that the ED results are provided).

Even though the difficulty of the amplitudes seems counter-intuitive, it may not be surprising if one recalls the path integral Monte Carlo. The sign problem in the path integral Monte Carlo implies that estimating the expectation values of observables $\mathrm{Tr}[A e^{-\beta H}] / \mathrm{Tr}[e^{-\beta H}]$ is difficult due to negative weights. As the amplitudes $|\psi_{\mathrm{GS}}(x)|^2$ are the expectation value of the observable $A = |x\rangle \langle x|$ when $\beta \to \infty$, the amplitudes $|\psi_{\mathrm{GS}}(x)|^2$ suffer from the sign problem when $H$ is non-stoquastic.

Still, this does not explain why learning amplitudes is difficult even for a supervised learning set-up where we already have values $|\psi_{\mathrm{GS}}(x)|^2$. A partial answer to this question can be found in Ref. [24]. Under common assumptions of the computational complexity (that the polynomial hierarchy does not collapse), Ref. [24] showed that computing the coefficients in the computational ba-

sis of a certain quantum state $[\Psi_{\mathrm{GWD}}(x)]$ is impossible in a polynomial time even with an exponential time of pre-computation (that may include training and computing the normalization constant of a neural network representation). As their argument relies on the difficulty of computing $|\Psi_{\mathrm{GWD}}(x)|^2$, the complexity is, in fact, from the learning amplitudes of quantum states. However, as they only considered a specific type of quantum states in 2D and do not give any argument on how errors scale (unlike the theory of DMRG which gives an upper bound of errors in terms of entanglement), further theoretical developments are essential to fully explain our results.

## VI. CONCLUSION

By way of example, we have classified the failure mechanism of the RBM Ansatz within the vQMC framework for one-dimensional spin systems. In particular, we have observed the following features of RBM variational Monte-Carlo for a class of one-dimentional XYZ-type Hamiltonians which exhibit a wide variety of stoquastic and non-stoquastic phases:

1. Complex RBMs with a constant hidden layer density ($\alpha$) faithfully represent the ground states of spin Hamiltonians that are phase connected to a stoquastic representation, or that can be transformed into such a Hamiltonian with local Pauli and phase-shift transformation.

2. There exists "deep non-stoquastic phases" that cannot be transformed into a stoquastic form using local (on-site) unitaries and are not phase connected to stoquastic Hamiltonians, and which cannot be efficiently represented by complex RBMs.

3. Sampling is stable along the learning path when the Hamiltonian is stoquastic or phase connected to a

stoquastic Hamiltonian. However, required number of samples to obtain true optima may scale poorly even in this case when the ground state distribution is heavy-tailed.

Most importantly, our observation 1 provides strong evidence suggesting that the RBM Ansatz can faithfully express the ground state of a non-stoquastic Hamiltonian when it is phase connected to a stoquastic Hamiltonian. This observation implies that it may be possible to solve a large number of non-stoquastic Hamiltonians using the RBM Ansatz, significantly expanding the reach of vQMC.

On the other hand, the second observation suggests that a fundamental difficulty may exist when solving a Hamiltoninan within a phase that is separated from any stoquastic Hamiltonian. As studies already have found several phases that cannot be annealed into stoquastic Hamiltonians [57–59], we expect that such systems are challenging for neural quantum states. Moreover, by carefully extending the supervised learning set-up introduced in Refs. [17, 54], we have further demonstrated that the difficulty in representing quantum states using a neural network is originated not only from their sign structure but also from the amplitudes. Even though this may sound counter-intuitive, but our result is consistent with that from the computational complexity theory [24].

Nevertheless, there is a caveat on our "deep non-stoquastic phases" as they rely rather on a conventional concept of phases (phase connectivity is restricted within a given *parameterized* Hamiltonian). In contrast, a modern language of one-dimensional phases allows any constant-depth local unitary transformations that preserve a symmetry between ground states, i.e. two Hamiltonians with the ground states $|\psi_1\rangle$ and $|\psi_2\rangle$ are in the same phase if there is a symmetry preserving unitary operator $U_{\mathrm{sym}}$ such that $|\psi_1\rangle = U_{\mathrm{sym}}|\psi_2\rangle$ and can be decomposed into a constant depth circuit consists of local gates [50–52]. One may possibly interpret our results using symmetry protected phases. Recently, Ref. [49] reported results on a related problem when a ground state can be transformed into a positive form under a unitary operator with a certain symmetry. However, as the result there is limited to ground states with hidden orders whereas our results suggest that ground states in phase $\Lambda$ of the twisted XYZ model suffer from the sign problem even though it is conventionally ordered, a further study is required to understand how phases of a many-body system interplay with a stoquasticity more deeply.

We also note that the neural networks we have used in this paper have $\leq 10^4$ parameters. Although this value is comparable to neural quantum states considered for the vQMC, modern machine learning applications use neural networks with several millions to billions of parameters. We open a possibility that such a huge network may sufficiently mitigate the *expressivity* problem we have observed in this paper. However, the main obstacle in using huge networks for the vQMC is the computational overhead of the SR which requires $O(|\theta|^{2-3})$ of operations for each step. Thus a better optimization algorithm would be required.

Finally, we also have found that the number of samples to solve the ground state may scale poorly even when the system is stoquastic. This type of difficulty is known from quantum chemical systems [53] but has not been discussed in solving many-body Hamiltonians. We still do not exclude the possibility that an efficient sampling algorithm that converges the network in a reasonable number of epochs may exist even in this case.

## ACKNOWLEDGEMENT

The authors thank Prof. Simon Trebst, Dr. Ciarán Hickey, and Dr. Markus Schmitt for helpful discussions. This project was funded by the Deutsche Forschungsgemeinschaft under Germany's Excellence Strategy - Cluster of Excellence Matter and Light for Quantum Computing (ML4Q) EXC 2004/1 - 390534769 and within the CRC network TR 183 (project grant 277101999) as part of project B01. The numerical simulations were performed on the JUWELS and JUWELS Booster clusters at the Forschungszentrum Juelich. This work presented in the manuscript was completed while both authors were at the University of Cologne. Source code used in this paper is available at Ref. [60].

## Appendix A: Training complex RBMs

### 1. Initialization

We initialize the parameters $\theta$ of the complex RBM using samples from the normal distribution $\mathcal{N}(0, \sigma^2)$. We typically use $\sigma = 10^{-3}$ but $\sigma = 10^{-2}$ also have reported almost the same converged energies in most of simulations.

### 2. Sampling

We have used the Metropolis-Hastings algorithm with the parallel tempering method to sample from the complex RBMs. Our set-up follows that of Ref. [61], which we summarize in this Appendix briefly.

To sample from the complex RBM $\psi_\theta(x)$, we first initialize the configuration $x$. For each step, we choose a new configuration $x'$ following a certain update rule. When the system has the $U(1)$ symmetry, we exchange $x_i$ and $x_j$ for randomly chosen $i$ and $j$. Otherwise, we choose $i \in [1, \cdots, N]$ randomly and flip $x_i$, i.e. $x'_j = x_j - 2x_j \delta_{j,i}$. Then the Metropolis-Hastings algorithm accepts the new configuration with probability

$$p_{\mathrm{accept}} = \min\left\{1, \left|\frac{\psi_\theta(x')}{\psi_\theta(x)}\right|^2\right\}. \tag{A1}$$

After defining a (complex) quasi-energy $E(x;\theta) = \sum_{i=1}^{N} a_i x_i + \sum_{j=1}^{M} \log[\cosh(\chi_j)]$ where $\chi_j = \sum_i w_{ij} x_i + b_j$, we can write

$$\psi_\theta(x) = e^{a \cdot x} \prod_j 2\cosh(\chi_j) = 2^M e^{E(x;\theta)} \qquad (A2)$$

which follows from Eq. 2. Then the update probability is simply written as $|\psi_\theta(x')/\psi_\theta(x)|^2 = e^{2\Re[E(x';\theta) - E(x;\theta)]}$. We call $N$ sequential MCMC steps a "sweep".

We also use this equation to generate a Markov chain with a given temperature $\beta$ that is for the probability distribution given as $P(x) \propto |\psi_\theta(x)|^{2\beta}$. One only needs to change the update probability to $e^{2\beta\Re[E(x';\theta) - E(x;\theta)]}$. We use total 16 chains with the equal spaced inverse temperatures $\beta = 1/16$ to 1 for parallel tempering. Let $i$-th chain ($i \in [1, \cdots, 16]$) has $\beta = i/16$ and the configuration $x^{(i)}$. After sweeping all chains, we mix them as follows: For all odd $i \in [1, 3, \cdots, 15]$, exchange $x^{(i)}$ and $x^{(i+1)}$ with probability

$$p_{\text{accept}} = \min\Big(1, \exp\big\{2(\beta_i - \beta_{i+1}) \times$$
$$\Re[E(x^{(i)};\theta) - E(x^{(i+1)};\theta)]\big\}\Big) \qquad (A3)$$

and do the same for all even $i \in [2, 4, \cdots, 14]$.

For the XXZ and the $J_1$-$J_2$ model in Fig. 1, we have used $|\theta|$ (the number of parameters of the RBM) number of samples per each epoch. For Fig. 4, we have used $|\theta|$ number of samples for phase I whereas $16|\theta|$ number of samples is used for phase $\Lambda$ and II (see also Fig. 5).

### 3. Hyperparameters

We use the the SR to train the complex RBMs (see also Sec. II). We typically use the learning rate $\eta = 0.02$ and the regularization constant $\lambda_n = \max\{1.0 \times 0.9^n, 10^{-3}\}$. For Phase II of the twisted-XYZ model, we use running averages of the gradient and the Fisher matrix with $\beta_1 = \beta_2 = 0.9$ to update parameters as it is slightly better performing than the SR without momentums. We train the network for $2500 - 3000$ epochs where we always have observed convergence besides Phase II of the twisted XYZ model (see Appendix G).

### Appendix B: The XXZ model with exact sampler

We have shown in the main text that the XXZ model without the sign rule suffers from the *sampling* problem in the antiferromagnetic phase $\Delta > 1.0$. In this Appendix, we use the exact sampler to show that this is *not* caused by temporal correlations of the MCMC.

Our exact sampler works as follows. For a given domain of configurations $\mathcal{D} = \{x_i\}$ (thus $|\mathcal{D}| = 2^N$ or $\binom{N}{N/2}$ depending on the symmetry of the system), we first compute and save $p(x) = |\psi_\theta(x)|^2$ for all $x \in \mathcal{D}$. After this

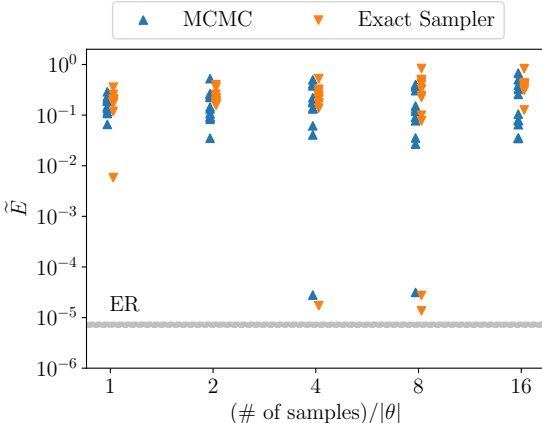

FIG. 9. For the XXZ model with $\Delta = 1.5$ and the system size $N = 20$, we show converged normalized energies of the complex RBM wavefunctions with $\alpha = M/N = 3$ when the SR is used with different sampling methods (MCMC and the exact sampler). Values of the $x$-axis indicate the number of samples we have used to train the RBM for each epoch divided by the number of parameters $|\theta| = \alpha(N+1) + \alpha N^2$. For each sampling method and the number of samples, we have run 12 randomly initialized instances. The shaded region (near $\widetilde{E} \approx 10^{-5}$) indicate the range of converged energies form the ER.

precomputation (the complexity of which is $O(|\mathcal{D}|)$), each sample is taken by finding $k$ such that $\sum_{i<k} p(x_i) < p$ and $\sum_{i \le k} p(x_i) > p$ where $p$ is taken from the uniform distribution of $\mathcal{U}_{[0,1]}$. Then $x_k$ is a valid sample from the distribution. As we implement this procedure with a binary search, time complexity of obtaining each sample is $O(\log|\mathcal{D}|) \sim N$. Then one may train the RBM using the SR with samples drawn from the exact sampler. We note that the ER can be regarded as a limiting case of the SR with the exact sampler where the number of samples per each epoch becomes infinite.

Using the XXZ model without the sign rule where the *sampling* problem appeared, we compare the converged normalized energies from the SR with the usual MCMC and the exact sampler in Fig. 9. The size of the system $N = 20$ and $\Delta = 1.5$ are used. The results clearly demonstrate that the SR with the exact sampler is not particularly better than the SR with MCMC, which confirms that the poor converged energies from the SR is not from the ergodicity issue.

### Appendix C: Non-stoquasticity of the the Heisenberg XYZ models

In this Appendix, we present an algorithm introduced by Klassen and Terhal [34] that determines (non-)stoquasticity of spin chains, and apply it to the $J_1$-$J_2$ and twisted XYZ models studied in the main text.

First, we interpret the XYZ type Hamiltonian as a graph with matrix-valued edges. For each Hamil-

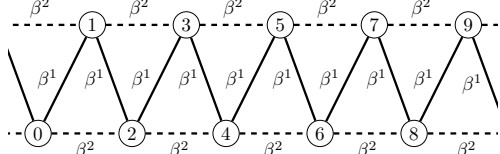

FIG. 10. Graph with matrix-valued edges describing translational invariant models with the nearest-neighbor and the next-nearest-neighbor couplings.

tonian term between vertices $(i,j)$ given by $H_{i,j} = \beta_{ij}^x \sigma_i^x \sigma_j^x + \beta_{ij}^y \sigma_i^y \sigma_j^y + \beta_{ij}^z \sigma_i^z \sigma_j^z$ we assign a matrix $\beta_{ij} = \mathrm{diag}(\beta_{ij}^x, \beta_{ij}^y, \beta_{ij}^z)$ to edge $(i,j)$. For a many-body spin-1/2 Hamiltonian consisting of two-site terms, it is known that the Hamiltonian is stoquastic if and only if all terms are also stoquastic. Using the matrix $\beta$, the term $H_{i,j}$ is stoquastic when $\beta_{ij}^x \leq -|\beta_{ij}^y|$. In addition, if there is a local (on-site) basis transformation that converts the Hamiltonian into a stoquastic form, it must act as signed permutations to each vertex, i.e. transform the matrix $\beta_{ij}$ to $\widetilde{\beta}_{ij} = \Pi_i \beta_{ij} \Pi_j^T$ where $\Pi_i = R_i \Pi_i$, $R_i = \mathrm{diag}(\pm 1, \pm 1, \pm 1)$ is a sign matrix and $\Pi_i \in S_3$ is a permutation (Lemma 22 of Ref. [34]). One may note that for signed permutations, the transformed Hamiltonian is still XYZ-type ($\widetilde{\beta}_{ij}$ are diagonal). Then we can formally write the problem as: find a set of signed permutations $\{\Pi_i = R_i \Pi_i\}$ that makes the transformed matrix $\widetilde{\beta}_{ij} = \Pi_i \beta_{ij} \Pi_j^T$ satisfy $\widetilde{\beta}_{ij}^x \leq -|\widetilde{\beta}_{ij}^y|$ for all edges $(i,j)$. The algorithm solves this by separating the permutation and the sign parts.

1. First, check whether there is a set of permutations $\{\Pi_i \in S_3\}$ that make $|\widetilde{\beta}_{ij}^x| \geq |\widetilde{\beta}_{ij}^y|$. This step can be done efficiently utilizing the fact that permutations $\Pi_i$ and $\Pi_j$ must be the same if the rank of $\beta_{ij}$ is $\geq 2$ (Lemma 23 of Ref. [34]).

2. Second, for possible permutations obtained from the above step, check if there is a possible set of signs $\{R_i\}$ that makes $\widetilde{\beta}_{ij}^x$ negative. This is reduced to a problem of deciding whether to apply $\mathrm{diag}(-1, 1, 1)$ or not to each vertex (Lemma 27 of Ref. [34]), which is equivalent to solving the frustration condition of classical Ising models that can be solved in a polynomial time.

We refer to the original work Ref. [34] for detailed steps. We also note that the algorithm finds a solution *if and only if* such a transformation exists.

We depict the interaction graph for translational invariant models with the next-nearest-neighbor couplings in Fig. 10. For this type of graph, the second problem (finding the possible signs) is much easier to solve. In our graph, applying $\mathrm{diag}(-1, 1, 1)$ to all even sites flips the sign of $\beta_1^x$. However, such a sign rule does not exist for $\beta_2^x$. Thus after the permutation, $\widetilde{\beta}_2^x$ must be negative whereas the sign of $\widetilde{\beta}_1^x$ for the nearest-neighbor couplings

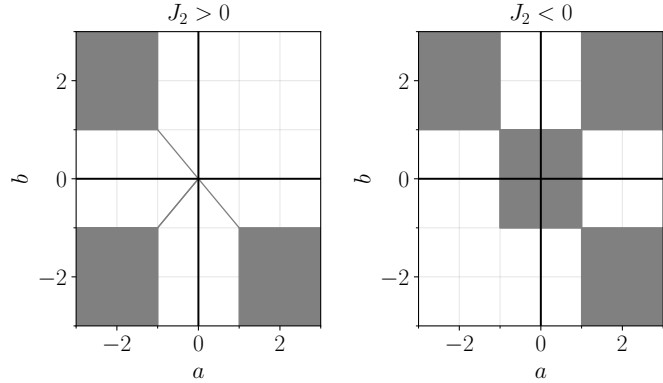

FIG. 11. Parameter regions that the twisted XYZ model can be transformed into a stoquastic form by on-site unitary gates.

is free as we can always make it negative by applying this sign rule.

### 1. Non-stoquasticity of the $J_1$-$J_2$ model

We can directly apply the algorithm described above to the $J_1$-$J_2$ model. For the $J_1$-$J_2$ model we have studied in the main text, we have $\beta$ matrices:

$$\beta_1 = J_1 \begin{pmatrix} 1 & 0 & 0 \\ 0 & 1 & 0 \\ 0 & 0 & 1 \end{pmatrix}, \qquad \beta_2 = J_2 \begin{pmatrix} 1 & 0 & 0 \\ 0 & 1 & 0 \\ 0 & 0 & 1 \end{pmatrix}. \qquad \text{(C1)}$$

As a permutation does not change the matrices, we only need to consider the sign rule which readily implies the Hamiltonian is stoquastic only when $J_2 \leq 0$. Still, it is known that the $J_1$-$J_2$ model can be transformed into a stoquastic form using two-qubit operations [62].

In addition to the non-stoquasticity of the Hamiltonian, one can further prove that ground states in the gapped phase of this model (when $J_2 > J_2^*$) indeed cannot be transformed into a positive form easily. Reference [49] states that a phase that is short-ranged entangled and has a string order suffers a symmetry protected sign problem – the transformed ground state in the computational basis cannot be positive (there exists $x$ such that $\langle x|U_{\mathrm{sym}}|\psi_{\mathrm{GS}}\rangle \notin \mathbb{R}_{\geq 0}$) for all symmetry protecting unitary operators $U_{\mathrm{sym}}$. We can directly apply this result to our case as the phase of the $J_1$-$J_2$ model when $J_2 > J_2^*$ satisfies this condition (see e.g. Ref. [63]). As the symmetric group is $G = SO(3)$ in the $J_1$-$J_2$ model, this implies that a translational invariant gate $U^{\otimes N}$ which transforms the ground state into a positive form does not exist. This further supports a relation between the expressive power of the RBM and stoquasticity of the Hamiltonian. Interestingly, at the Majumdar-Ghosh point ($J_2 = 0.5$), a non-translational invariant unitary gate $\otimes_i \sigma_{2i}^z$ transforms the ground state into a positive form yet the transformed Hamiltonian is still non-stoquastic.

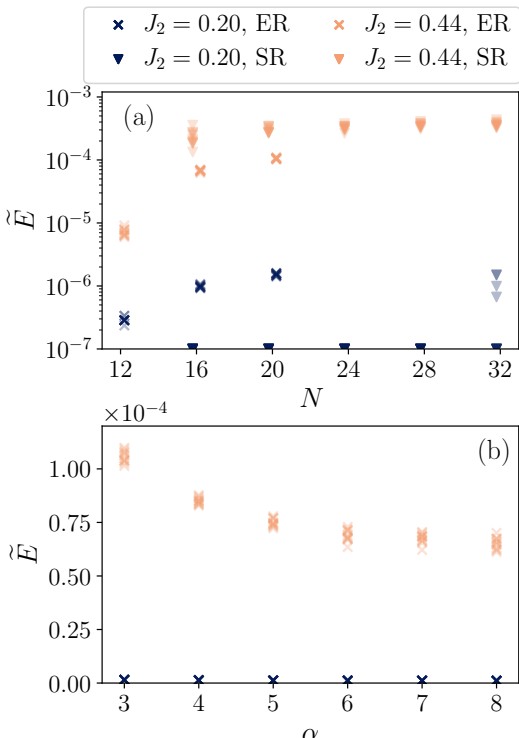

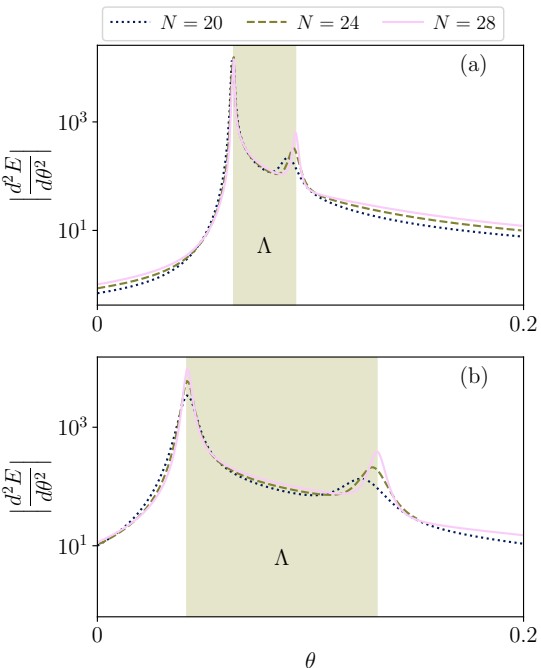

FIG. 12. (a) Converged normalized energies $\widetilde{E} = (E_{\mathrm{RBM}} - E_{\mathrm{ED}})/E_{\mathrm{ED}}$ from the ER and the SR as a function of different system size $N$ for the $J_1$-$J_2$ model with $J_2 = 0.2$ and $0.44$. We applied the sign rule and the ratio between hidden and visible units $\alpha = 3$ is used. (b) Converged normalized energies as a function of the number of hidden units $\alpha = M/N$ when $N = 20$.

### 2. Non-stoquasticity of the twisted XYZ model

We apply the same algorithm to the twisted XYZ model and determine the parameter regions where the twisted XYZ model can be locally transformed into stoquastic form.

For the twisted XYZ model, the $\beta$ matrices are given as

$$\beta_1 = J_1 \begin{pmatrix} a & 0 & 0 \\ 0 & b & 0 \\ 0 & 0 & 1 \end{pmatrix}, \qquad \beta_2 = J_2 \begin{pmatrix} b & 0 & 0 \\ 0 & a & 0 \\ 0 & 0 & 1 \end{pmatrix}. \qquad \text{(C2)}$$

The model is trivially stoquastic when $a = b = 0$. If any of $a$ or $b$ is non-zero, we see the ranks of the matrices are $\geq 2$. Thus all permutations acting on each vertex must be equal, i.e. $\Pi_i = \Pi$. After this simplification, we can just apply each element $\Pi \in S_3$ and see whether there is a set of sign matrices that satisfies the condition.

For our model, it is not difficult to obtain the condi-

FIG. 13. Second derivatives of the ground state energies along two paths (a) $P_1$ and (b) $P_2$ depicted in Fig. 6 of the main text. Path $P_1$ is from $(a, b) = (0.7, 1.2)$ to $(0.9, 1.4)$ that is a part of $\overline{AE}$ in Fig. 2 that we simulated vQMC. Path $P_2$ is from $(0.55, 1.45)$ to $(0.75, 1.65)$.

tions below:

$$\begin{aligned} \Pi = () && |a| = |b| \cap J_2 b \leq 0 && \text{(C3)} \\ \Pi = (1, 2) && |b| = |a| \cap J_2 a \leq 0 && \text{(C4)} \\ \Pi = (2, 3) && |a| \geq 1 \cap |b| \geq 1 \cap J_2 b \leq 0 && \text{(C5)} \\ \Pi = (1, 3) && 1 \geq |b| \cap 1 \geq |a| \cap J_2 \leq 0 && \text{(C6)} \\ \Pi = (1, 2, 3) && 1 \geq |a| \cap 1 \geq |b| \cap J_2 \leq 0 && \text{(C7)} \\ \Pi = (1, 3, 2) && |b| \geq 1 \cap |a| \geq 1 \cap J_2 a \leq 0 && \text{(C8)} \end{aligned}$$

which is depicted in Fig. 11.

When $J_1 = J_2 = -1$ and $a, b \geq 0$, that we considered in the main text, we see that $\Pi = (2, 3)$ and $\Pi = (1, 3)$ are the transformations that make the Hamiltonian stoquastic for $a, b \geq 1$ and $a, b \leq 1$, respectively. As the elements of the transformed matrices $\widetilde{\beta}_1 = \Pi \beta_1 \Pi^T$, $\widetilde{\beta}_2 = \Pi \beta_2 \Pi^T$ are already negative, additional application of the sign rule is not required.

### Appendix D: Expressive power of the RBM for the $J_1$-$J_2$ model

In the main text, we have observed that the errors from the $J_1$-$J_2$ model when $J_2 \in (J_2^*, 0.5)$ are large even when the ER is used and the sign rule is applied. In this section, we investigate the errors in more detail using the RBM with different system sizes and number of hidden units.

For $J_2 = 0.20$ and 0.44, we plot converged normalized energies for different system sizes $N$ and values of $\alpha = M/N$ in Fig. 12. Figure 12(a) shows that both the ER and the SR find the ground state when $J_2 = 0.2$. We also observe the MCMC samples bias the energy a little toward the ground state and the energy and the statistical fluctuation get larger as $N$ increases in the SR case. However, one can easily deal with such a problem by carefully choosing the sample size and the number of sweeps between samples. In contrast, we see that the *expressivity* problem arises when $J_2 = 0.44$, which is more fundamental. In addition, even though the errors seem to remain constant with $N$ for the SR, we expect the obtained state to be moving away from the true ground state as $N$ increases because the normalized energy of the first excited state scales as $\sim 1/N$.

We additionally study how the error scales with the number of hidden units in Fig. 12(b). The results clearly show that increasing $\alpha = M/N$ only marginally reduces the error. We also note that the system at $J_2 = 0.44$ is gapped thus the error from $\alpha = 8$ is still far from the error we expect from other methods e.g. density matrix renormalization group. This supports our argument that the RBM has a difficulty in representing the ground state of Hamiltonians that is deep in non-stoquastic phase.

## Appendix E: Phases of the twisted XYZ model

In the main text, we summarized the phases of the twisted XYZ model. Here, we investigate the phases more closely and locate phase transition points along the path $\overline{AE}$. Let us first consider the case when $a = b$ that recovers the XXZ model with the next-nearest-neighbor couplings. After setting $J_1 = J_2 = -1$, the Hamiltonian becomes

$$H_{tXYZ} = \sum_i -(C\sigma_i^x\sigma_{i+1}^x + C\sigma_i^y\sigma_{i+1}^y + \sigma_i^z\sigma_{i+1}^z)$$
$$- (C\sigma_i^x\sigma_{i+2}^x + C\sigma_i^y\sigma_{i+2}^y + \sigma_i^z\sigma_{i+2}^z) \quad \text{(E1)}$$

where we set $a = b = C$. Applying Pauli-$Z$ gates to all even sites (the sign rule) yields

$$H_{tXYZ}^* = C\sum_i (\sigma_i^x\sigma_{i+1}^x + \sigma_i^y\sigma_{i+1}^y - 1/C\sigma_i^z\sigma_{i+1}^z)$$
$$- (\sigma_i^x\sigma_{i+2}^x + \sigma_i^y\sigma_{i+2}^y + 1/C\sigma_i^z\sigma_{i+2}^z). \quad \text{(E2)}$$

The Hamiltonian has $U(1)$ symmetry, so it preserves the total magnetization $J_z = \sum_i \sigma_i^z$. When $C < 1$, it is not difficult to see that the nearest-neighbor and the next-nearest-neighbor couplings both prefer the aligned states, which suggests that $|0\rangle^{\otimes N}$ and $|1\rangle^{\otimes N}$ are the degenerated ground sates. This implies that the $\mathbb{Z}_2$ symmetry along the $z$-axis is broken. In contrast, the Hamiltonian with $C > 1$ prefers $J_z = 0$ thus $\mathbb{Z}_2$ symmetry $\sigma_z \leftrightarrow -\sigma_z$ is restored.

Fig. 2 in the main text shows how the phases extend off $a = b$. Especially, there are two phase transitions along

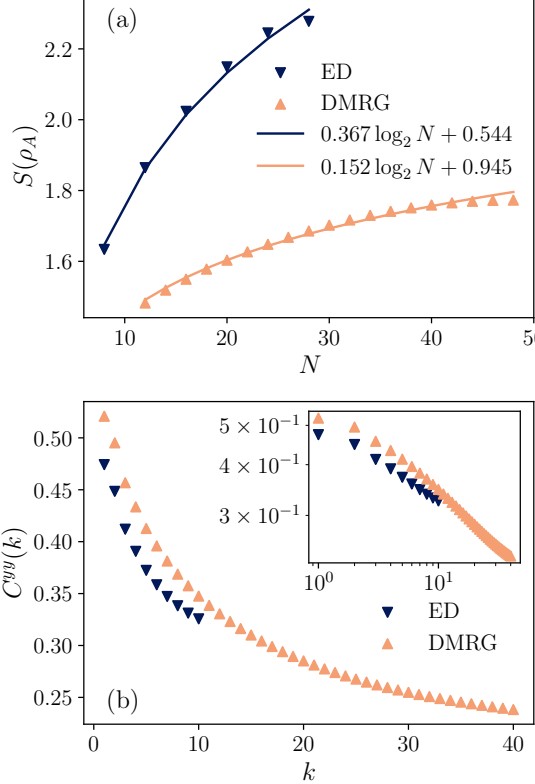

FIG. 14. (a) Scaling of entanglement entropy at $C_1$ from the ED (periodic boundary condition) up to $N = 28$ and the DMRG (open boundary condition) up to $N = 48$ and (b) the spin-spin correlation along the $y$-axis as a function of distance $k$ at $E = (1.25, 1.75)$ from the ED with $N = 28$ and the DMRG with $N = 48$. Entanglement entropy is obtained after dividing the system into two equal sized subsystems $A$ and $B$. Thus $\rho_A = \text{Tr}_B[|\text{GS}\rangle\langle\text{GS}|]$ and $S(\rho_A) = -\text{Tr}[\rho_A \log(\rho_A)]$ where $|GS\rangle$ is the ground state of the Hamiltonian at the given point. Inset of (b) shows the same data in log-log scale.

the line segment from $A = (0.25, 0.75)$ to $E = (1.25, 1.75)$ that we have simulated vQMC. To locate the second transition point, we calculate $d^2E(\theta)/d\theta^2$ where $E(\theta)$ is the ground state energy of $H_{tXYZ}$ where $\theta$ parameterizes the path. We use paths $P_1$ and $P_2$ depicted in Fig. 6 of the main text. As path $P_1$ is a subset of $\overline{AE}$ we simulated the vQMC, we can locate the point $C_2$ in Fig. 2 using the result. Path $P_2$ is additionally considered to confirm that our phase diagram Fig. 2 indeed captures phase $\Lambda$ correctly. To be precise, we set $(a, b) = (0.7, 1.2) + \theta(1, 1)$ for $P_1$ and $(a, b) = (0.55, 1.45) + \theta(1, 1)$ for $P_2$. The results from $P_1$ and $P_2$ are shown in Fig. 13(a) and (b), respectively. We see that there are two local maxima in the second derivatives along each path and the distance between two maxima increases as the path is moving away from the $a = b = 1$ point. This confirms that an intermediate phase $\Lambda$ depicted in Fig. 2 is correct.

We additionally confirm some other properties of the model using the DMRG with the open boundary condi-

tion. In the main text, we have argued that the phase transition at $C_1$ is the second order. We calculate the entanglement entropy at $C_1$ by taking the entropy of a subsystem after dividing the system of size $N$ to two equal-sized subsystems $A$ and $B$ in Fig. 14(a). As we have used the periodic and open boundary conditions for the ED and DMRG, respectively, the coefficients $0.367 \sim 1/3$ and $0.152 \sim 1/6$ are close to what one expects from the conformal field theory [64]. We also calculate the spin-spin correlation along the $y$-axis $C^{yy}(k) = \sum_{i=1}^{N} \langle \sigma_y^i \sigma_y^{i+k} \rangle / N$ at point $E = (1.25, 1.75)$ of the twisted-XYZ model in Fig. 14(b). The result supports that the correlation function decays polynomially in Phase II.

### Appendix F: Scaling of converged energies in Phase II of the twisted XYZ model

In the main text, the converged energies along the path $\overline{C_2 D}$ with the ER and the SR have shown a different shape when $H_{\text{tXYZ}}^{\diamond}$ is used. In this Appendix we show that this is a finite size effect.

To study this case more carefully, we simulate the ER

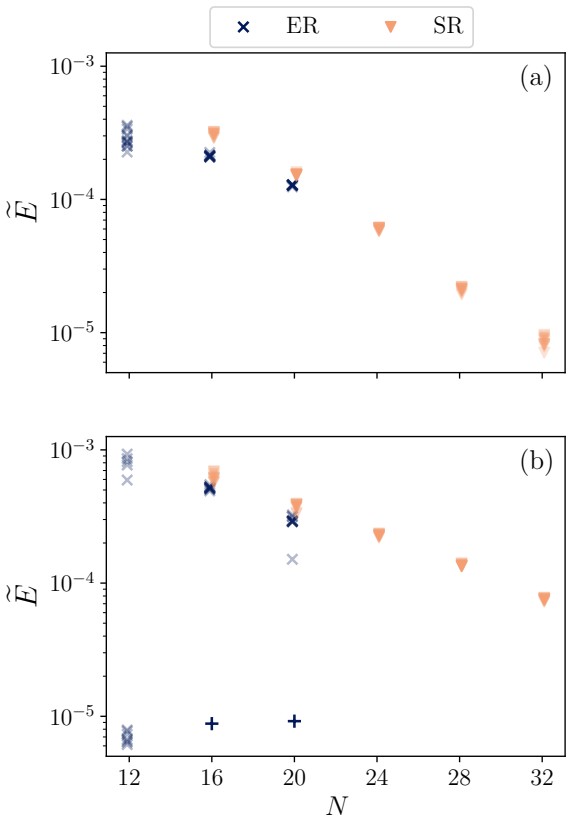

FIG. 15. Converged normalized energies for $H_{\text{tXYZ}}^{\diamond}$ with parameters (a) $(a, b) = (0.81, 1.31)$ and (b) $(0.89, 1.39)$ using the ER and SR. Plus markers $(+)$ in (b) indicate converged energies from the Hamiltonian annealing.

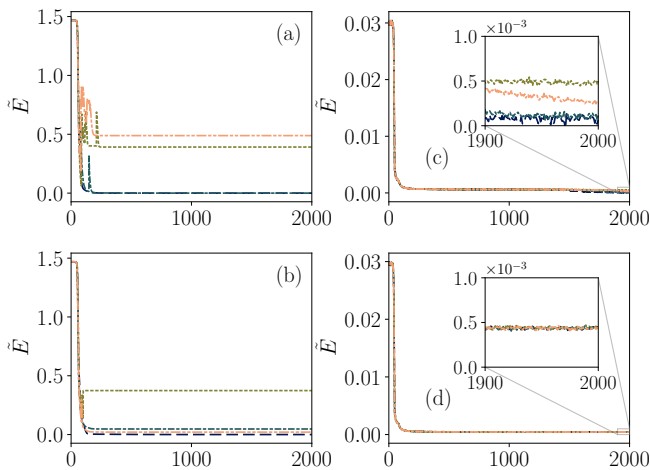

FIG. 16. Four randomly chosen learning curves from (a, b) the XXZ model at $\Delta = 1.5$ without the sign rule and (c, d) the twisted XYZ model deep in phase II when $(a, b) = (1.23, 1.73)$. The system size $N = 28$ is used for (a) and (c), and $N = 32$ is used for (b) and (d).

for system sizes $N = [12, 16, 20]$ and the SR for system sizes $N = [16, 20, 24, 28, 32]$ using the two points indicated by arrows in Fig. 4 which are $(a, b) = (0.81, 1.31)$ and $(0.89, 1.39)$. The results in Fig. 15 show that converged energies from the SR decrease exponentially with $N$ which suggests that the optimizing problem vanishes for large $N$ and we can solve the Hamiltonian using the SR.

One possible explanation for how such a good convergence is obtained for large $N$ is overparametrization. In classical ML, it is known that overparamterized networks optimize better [65]. Likewise, if the ground state is already sufficiently described by a small number of parameters, we expect that one can obtain a better convergence by increasing the network's parameter. As the number of parameters of our network increases quadratically as $\sim \alpha N^2$, this scenario is plausible if the number of parameters to describe the ground state scales slower than this. However, we leave a detailed investigation of this conjecture for future work.

### Appendix G: Sampling problems from non-stoquastic basis and from heavy tail distributions

In the main text, we have observed two different types of *sampling* problems. The first one appeared when we used a Hamiltonian in a non-stoquastic basis (e.g. the XXZ model in the antiferomagnetic phase without the sign rule). When this happened, converged energies of most of the SR instances are clustered far above the ground state energy $(\widetilde{E} > 10^{-2})$ and it persists regardless of the system size and the number of samples. We next observed a seemingly similar problem when solving

the twisted XYZ model in Phase II using the stoquastic version of the Hamiltonian. However, the converged energies in this case were much closer to the ground state energy ($\widetilde{E} < 10^{-3}$ for all instances) and increasing the number of samples helped for $N \leq 28$. In this Appendix, we compare the learning curves in both models and show that the problems indeed have different profiles.

For comparison, we use the XXZ model with $\Delta = 1.5$ and the twisted XYZ model with $(a, b) = (1.23, 1.73)$. We plot 4 randomly chosen learning curves when $N = 28$ and $N = 32$ for both models in Fig. 16. One can easily distinguish the learning curves from the XXZ model without the sign rule Fig. 16(a), (b) and the twisted XYZ model with heavy tail distributions (c), (d). Specifically, Fig. 16(a) and (b) clearly show that the *sampling* problem enters in the middle of learning process and ruins the learning process when we use a non-stoquastic basis. In contrast, the learning curves shown in Fig. 16(c) and (d) first approach local minima that is near $\widetilde{E} \approx 0.5 \times 10^{-3}$ and stay there for more than 1000 epochs. After that, some of the instances succeed in converging to better minima when $N = 28$. This behavior has also been seen in quantum chemical Hamiltonians [53]. These examples show that the origin of the two sampling problems are crucially different.

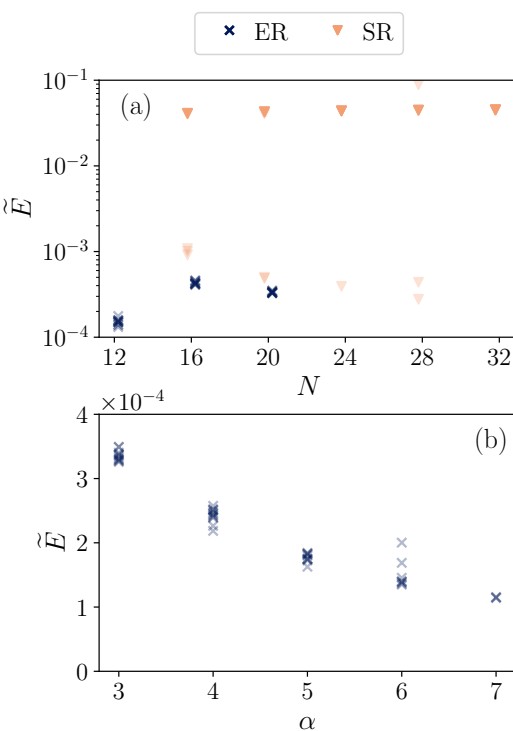

FIG. 17. Converged normalized energies of the twisted XYZ model at point $(a, b) = (0.4, 3.0)$ which is deep in phase $\Lambda$. The transformed Hamiltonian $H_{\text{tXYZ}}^{\star}$ is used. We plot (a) the results from the ER and SR as a function of $N$ when $\alpha = 3$ and (b) the ER result as a function of $\alpha$ when $N = 20$.

# Appendix H: Expressive power of the RBM for phase $\Lambda$ of the twisted XYZ model

In Sec. IV B 5, we have seen that errors from the ER increases as the parameter moves deeper in phase $\Lambda$ of the twisted XYZ model. In this Appendix, we study a scaling behavior of errors when the Hamiltonian parameter is deep in phase $\Lambda$. Especially, we use the point $K$ in Fig. 6 which is $(a, b) = (0.4, 3.0)$ and the Hamiltonian $H_{\text{tXYZ}}^{\star}$ which reported the best converged energies from the ER.

In contrast to the deep non-stoquastic phase of the $J_1$-$J_2$ model that we studied in Appendix D, our SR results in Fig. 17(a) suffer from the *sampling* problem. Still, the normalized energies seem to converge to a positive value $> 10^{-4}$ with $N$ even when we take the best results for each $N$. On the other hand, Fig. 17(b) shows the converged energies from the ER for varying $\alpha = M/N$ when $N = 20$. As in the $J_1$-$J_2$ model case [Fig. 12(b)], the improvements are getting marginal as $\alpha$ increases. From these observations, we confirm that the RBM does not represent the ground states in phase $\Lambda$ faithfully.

# Appendix I: Supervised learning set-ups

In Sec. V, we have studied expressivity of neural networks for reproducing amplitudes and signs of quantum states. We describe the details of our supervised learning set-ups in this Appendix.

## 1. Activation functions

After the convolutional layers (Conv-1 and Conv-2), we apply even and odd activation functions, respectively, to preserve the $\mathbb{Z}_2$ symmetry. For an even activation function, we use the cosine activation $\cos(\pi x)$ following Ref. [54]. For an odd activation function, we choose a "leaky" version of hard hyperbolic tangent given as

$$\text{LeakyHardTanh}(x) = \begin{cases} s(x-1) + 1 & \text{if } x \geq 1 \\ x & \text{if } |x| < 1 \\ -s(x+1) - 1 & \text{otherwise} \end{cases} \quad \text{(I1)}$$

where $s$ is the slope outside of $|x| \geq 1$. We use $s = 0.01$ in our simulations.

## 2. Initialization

We have mainly used `golot_normal` initialization suggested in Ref. [66] and implemented in `JAX`, as they have outperformed other initializers (e.g. `lecun_normal` and `he_normal`) for a various range of hyperparmeters.

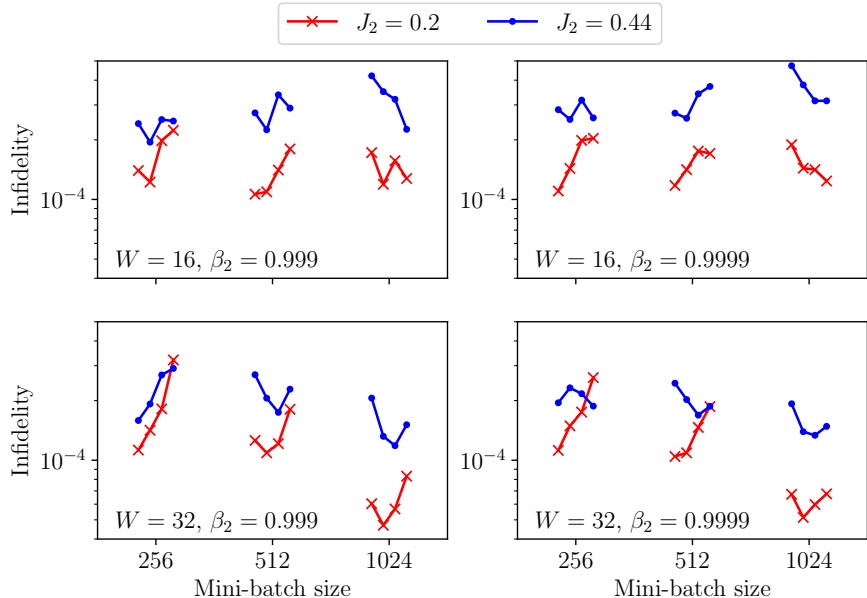

FIG. 18. Converged infidelity $1 - F$ of the amplitude network with different choices of hyperparameters of the natural gradient descent. For the one-dimensional $J_1$-$J_2$ model with $J_2 = 0.2$ and $0.44$, we have trained the network to reproduce the amplitudes of true ground states. For each mini-batch size, four different learning rates $\eta \in [5.0 \times 10^{-5}, 1.0 \times 10^{-4}, 2.0 \times 10^{-4}, 4.0 \times 10^{-4}]$ (from left to right) are used.

### 3. Choosing hyperparameters

#### a. Hyperparameters for the amplitude networks

We have used the natural gradient descent to train the amplitude networks. Hyperparameters are the learning rate $\eta$, the momentum for the gradient $\beta_1$, the momentum for the Fisher matrix $\beta_2$, and the mini-batch size. For all simulations, we have fixed $\beta_1 = 0.9$ as this is the default value for most of optimization algorithms implemented in major machine learning frameworks (e.g. `TensorFlow`, `PyTorch`, and `JAX`).

Using the one dimensional $J_1$-$J_2$ model with the system size $N = 24$ and $J_2 \in [0.2, 0.44]$, we test the neural networks with width $W \in [16, 32]$ and different learning rate $\eta$, $\beta_2 \in [0.999, 0.9999]$, and the mini-batch size $\in [256, 512, 1024]$ and plot the converged infidelities in Fig. 18. Regardless of the mini-batch size, the neural network have seen $\approx 1.57 \times 10^8$ datapoints (the configuration $x$ and the corresponding amplitude $|\psi_{GS}(x)|^2$) where we have seen a sufficient convergence behavior. For example, we have trained the network for $153,600$ epochs when the mini-batch size is 1024. Based on this result, we choose the mini-batch size 1024, $\eta = 1.0 \times 10^{-4}$, $\beta_2 = 0.999$ for plots in the main text.

#### b. Hyperparameters for the sign networks

We have observed that Adam [42] already works great for training the sign structure, as in usual supervised

FIG. 19. Converged infidelity $1 - F$ of the sign network with different choices of hyperparameters of Adam optimizer. For each mini-batch size, we have used six different learning rates $\eta \in [6.25 \times 10^{-5}, 1.25 \times 10^{-4}, 2.5 \times 10^{-4}, 5.0 \times 10^{-4}, 1.0 \times 10^{-3}, 2.0 \times 10^{-3}]$ (from left to right).

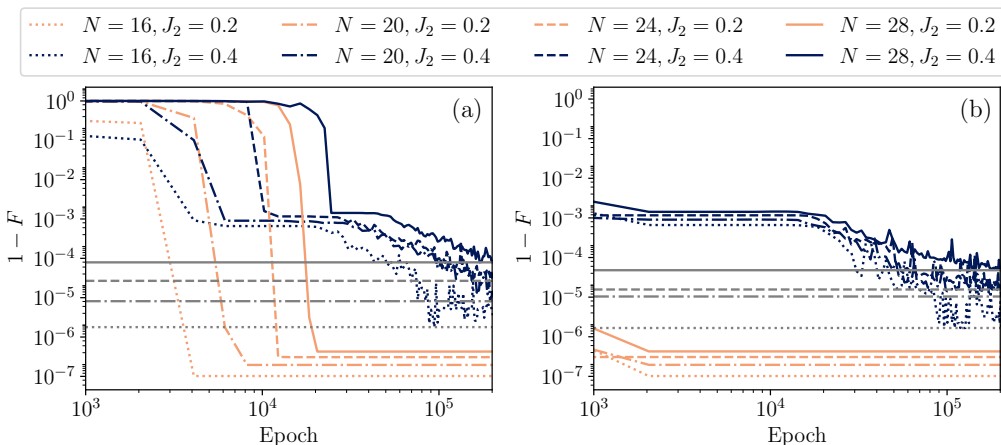

FIG. 20. Initial learning curves of the sign network from the one dimensional $J_1$-$J_2$ model (a) without and (b) with the sign rule. Grey horizontal lines indicate the converged infidelities when $J_2 = 0.4$. Imposing the sign rule gets rid of the initial warm-up stage of the learning but does not improve the converged infidelity.

learning processes in the classical ML set-ups (e.g. image classifications). Adam optimizer has the learning rate ($\eta$), momentum for the gradient and the gradient square ($\beta_1$ and $\beta_2$), and the constant for numerical stability ($\epsilon$) as hyperparameters. As the default values for $\beta_1$, $\beta_2$ and $\epsilon$ suggested in the original paper [42] work well for most of applications, we only tested the performance of Adam for varying learning rates $\eta$.

For mini-batch sizes $\in [32, 64, 128]$, we plot the converged infidelities using the one dimensional $J_1$-$J_2$ model in Fig. 19. Regardless of the mini-batch sizes, we feed $\approx 1.31 \times 10^8$ datapoints. Based on this result, we choose the mini-batch size 32 and the learning rate $\eta = 2.5 \times 10^{-5}$, which is the best performing for $W = 16$ and the second

best for $W = 32$, for plots in the main text.

### 4. Imposing the sign rule

In the main text, we have trained neural networks to reproduce the sign structure of the one dimensional $J_1$-$J_2$ model without imposing the sign rule. Here, we train network to reproduce the sign structure when the sign rule is imposed and compare the performance. We show initial learning curves and converged infidelities in Fig. 20. We see that initial warm-up stage of the learning disappears when the sign rule is imposed [Fig. 20(b)], but the converged infidelities are barely affected. This resembles the vQMC results with the complex RBM where imposing the sign rule does not improve the converged energies.

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
