# Peer review of "Expressive power of complex-valued restricted Boltzmann machines for solving non-stoquastic Hamiltonians"

_SciPost Physics_

## Round 2 · Referee Report · Anonymous · 2021-9-20

Strengths

1-comprehensive review of the role of stoquasticity in RBM-based variational Monte Carlo
2-identification of three distinct mechanisms for learning failure
3-techniques to distinguish these mechanisms in numerical experiments

Weaknesses

1-narrow focus on RBM wave functions
2-unconvincing discussion of the expressivity issue

Report

I thank the authors for carefully revising their manuscript and taking all points of my previous report on board. The technical discussion of the numerical experiments and their results are now quite comprehensible and complete.

The authors have added a new section (§5) to address the contradiction between their results (which suggest that the RBM ansatz is unable to represent the ground states of "deep non-stoquastic" Hamiltonians) and those of Ref. [17] (which show that shallow networks-albeit not RBMs-can represent such states to a high accuracy). The authors' view is that the standard of Ref. [17] for representing a ground state "accurately" is too lax, and a more stringent analysis of the same protocol distinguishes "extended stoquastic" and "deep non-stoquastic" phases. However, they fail to demonstrate this convincingly:

1-The authors highlight the sentence "expressibility of the ansätze is not an issue—we could achieve overlaps above 0.94 for all values of J2/J1" from Ref. [17] to dismiss their claim of accurately representing the ground state. It is worth pointing out that this overlap was achieved for the spin-1/2 kagome Heisenberg antiferromagnet, a notoriously difficult problem, where achieving such an overlap with the ground state on a reasonably large system with any variational method would be an unprecedented breakthrough. The authors' choice of problem, the J1-J2 chain, would be most analogous to the J1-J2 square lattice in Ref. [17], where that too achieves essentially perfect ground states.

2-The data presented in Fig. 7 don't show any clear signature of the BKT transition at J2/J1=0.24. The amplitude infidelity evolves smoothly in a factor-of-3 range; the sign infidelity rises from essentially 0 to a constant value well below the transition point. The most noticeable feature in both plots is not the transition, but the Majumdar-Ghosh point.

3-The ansätze used in this section are completely unrelated to RBMs: they contain two hidden layers and their activation functions are rather different from the coshes in an RBM. It is not clear (neither from this work nor the wider literature) whether RBMs could achieve the same high accuracy of representation in "deep non-stoquastic phases".

4-Out of the three Hamiltonians considered in other parts of the work, the authors chose the one for which every VMC protocol returned very good variational energies in all phases, thus a clear phase separation was never expected. (It is telling that the amplitude infidelity turned out orders of magnitude higher than the phase infidelity, contrary to virutally all experience with neural-network wave functions on frustrated Hamiltonians.) They would have had a much better chance with the J1-J2 XYZ chain.

Ironically, the conclusion sought by the authors can be read off much better from Fig. 2c of Ref. [17], which details results for the kagome lattice. The "expressibility" line (equivalent to the calculations here) falls noticeably below 1 in the "deep non-stoquastic phase", and the (second-order) transition into the same is clearly indicated by the line shape.
Even so, it is clear from the same figure that using only a subset of the computational basis in training leads to much worse results, most consistent with the "convergence" issue in the present paper's language.

In summary, the authors try and fail to substantiate the "expressivity" issue with RBM wave functions beyond §4; the arguments there remain sound and support the notion that RBMs (but not necessarily other networks) suffer from an expressivity issue. I would suggest that the authors rework §5 thoroughly to address the concerns above.

In light of the discussion above, however, it is unclear how important the expressivity issue is, even if it's there. Since the sampling issue is not studied in detail, and the convergence issue has already been addressed by Refs. [17,33], the only significant advance by this work is introducing a careful scheme to quantify and distinguish them. This falls short of the standards of SciPost Physics, so I recommend that the paper, after addressing my concerns, be published in SciPost Physics Core.

---

## Round 2 · Referee Report · Anonymous · 2021-9-29

Report

I thank the authors for improving the manuscript according to my and other referees previous comments, which I recommend for publication in SciPost Physics Core.

---

## Round 2 · Referee Report · Anonymous · 2021-10-4

Report

I thank the authors for providing a revised version of the manuscript.

I believe the revised version has significantly improved the quality of the presentation of the results and clarified several of the points that were raised by myself and by the other referees.

I believe that the comparison with existing literature, most notably for what concerns the new added section, is still slightly problematic. In this respect I would invite to follow the suggestions from the other referee on this point, that are quite precise and find me in full agreement.

Otherwise, I would recommend the work for publication in Scipost Physics Core.

Here there is a few final remarks on the authors reply:
* * *
5. "It is known that if one can sample from the exact distribution can obtain any quantum expectation value of a local observable to arbitrary precision"

Our reply: The argument does not work in our case as our observables include the gradient of energy and the Fisher matrix, which are essential for training. Those observables do not admit a geometrically local representation (see also footnote on page 8). As mentioned above and noted in Report 1, we thus expect the sampling problem by a heavy-tail distribution is robust.
* * *
The condition for expectation values to be estimated with polynomially scaling (in the number of samples) precision is just that the variance of the observable is finite. I was mentioning the case of local observables because in that case the analysis is easier and it is known that the variance is finite by construction. In the case of gradients/ Fisher, I am not aware of a fully analysis, but it is relatively easy to convince oneself that it is possible to find activation functions such that the variance of the gradients is also finite (actually, such that it is also bounded everywhere). This is not the case for complex-valued RBM (because the gradients of log(Psi) can be divergent, but this is just a specific architecture/ specific choice of ansatz. For other typically adopted parameterizations such as <Z|Psi>=A(Z)*exp(i*Phi(Z)), with A and Phi represented by neural networks with real parameters and standard activations, it is easy to show that the variance of the gradients are bounded.
That's why I believe the conclusions concerning the heavy tails are not really relevant in general, and their conclusion maybe limited to complex-valued RBMs.
* * *
6. "Again, sampling issues appear also in the classical domain and are strictly unrelated to stoquasticity." and 7. "It is not expected that all classical models are efficiently samplable (again, no sign problem involved, just the fact that if you have a glassy landscape it is hard to find/sample from the ground state for any known computational method to the humankind)."

Our reply: Of course, there are classical models (such as spin-glasses) that are hard to sample. However, we are solving a translational invariant short-range interacting stoquastic Hamiltonian, and it is unclear whether quantum Monte Carlo encounters such a difficulty. It is indeed a fundamental question discussed in e.g. [Bravyi and Terhal, SIAM J. Comput. Vol. 39, No 4, 1462 (2009)], [Hastings and Freedman, Quantum Info. & Comp. 13, 1038 (2013)], and [Jarret, Jordan and Lackey, Phys. Rev. A 94, 042318 (2016)]. In this respect, our observations "Sampling is stable along the learning path when the Hamiltonian is stoquastic or phase connected to a stoquastic Hamiltonian" and "one may encounter such a problem even in solving a one dimensional stoquastic system which one would expect…" are completely valid. In addition, as we have already stated multiple times, our objective is to find out when such a case arises.

8. "This is quite surprising, since the MCMC simply uses the ratio between two probability densities ..., which is sign invariant. "I would certainly not emphasize what is "surprising" or not, since this is again highly dependent on the taste of the reader. On the contrary, I would argue that any reader with a reasonable experience in classical Monte Carlo would not find surprising at all that, say, MCMC fails on the classical Ising model at low temperatures, despite "the absence of sign problem" there. In that context, would the authors draw the conclusion that "it is surprising that convergence is not observed, despite the fact that the Boltzmann weight has no sign problem"? if the answer to this question is no, then I would strongly suggest to entirely rethink the nature of their conclusions on this point.

Our reply: We feel that the Referee underappreciates the role of the sign rule in the vQMC mentioned in the manuscript. It is well known that the MCMC for the spin glass at the low temperature fails because of the complicated energy landscape, which clearly follows from the frustrated nature of the Hamiltonian (which defines the distribution). On the other hand, as we stated in the manuscript, the sampling problem for the Hamiltonian without the sign rule is nothing to do with the property of the quantum states (which defines the distribution). Thus the sign rule is completely irrelevant to whether one can sample from the distribution of a given quantum state or not. As we stated in the manuscript, it only changes the learning path the optimization algorithm takes. This demonstrates that the origin of the sampling problem here is completely different from that of the spin glass. We further support that this is irrelevant to the ergodicity problem as the MCMC for the spin glass system in the revised manuscript by adding the results from the exact sampler (Appendix B). Still, as our statement before may have induced some confusion, we clarified the statement. Now we explicitly say that this sampling problem is not caused by the ground state distribution but by the different learning paths taken by the original and the sign-transformed Hamiltonians (the first paragraph of page 8).
* * *
For both these points, again there is a misundarstanding. First of all, QMC would fail also on the Transverse-Field Ising model in the classical limit (translationally invariant, no disorder, no frustration), if not supplemented with moves that are non local in space and time. As I was pointing out, MCMC can fail in several ways when the moves give rise to long correlation times . This is a general phenomenon that is not related necessarily to the presence of glassy phases.
In any case, I appreciate the changes done by the authors in the manuscript that have clarified the original meaning of their sentence.
* * *
9. "In Fig 1 c it is argued that there is an expressive power issue because "some instances" have errors above 10^-4. This seems to me by far the weakest of the conclusions drawn by the authors, since there is a typical error of 10^-3 even in Fig. 1 (d), the sampled case".

Our reply: We don't get this point. When there is an expressivity issue, it is impossible that the sampled case gives better energy than the exact case. Moreover, we emphasize that we have supported our conclusion by showing the scaling behavior of errors in Appendices D and H.
* * *
As also put forward by other referees, the accuracy levels here are quite arbitary. My observation was that claiming that there is an expressivity problem in cases where the errors are smaller than 10^-4 seems really too strict as a criterion and far from any realistic application of the variational method to problems in many body physics.

---

## Round 2 · Author Response

Dear Editor,

thank you for addressing our manuscript scipost_202102_00010v1. As we have clarified all concerns from the Reviewers, we believe our manuscript is now ready for publication in SciPost Physics.

Best regards,
Chae-Yeun Park and Michael Kastoryano

---

## Round 2 · List of Changes

1. Title has changed.
2. The introduction is expanded to clarify that our main study objective is to determine which Hamiltonians the neural quantum states based vQMC encounter difficulties.
3. Section 5 is added to resolve the discrepancy between previous works and ours in the expressivity issue.
4. Appendix A that discusses the parallel tempering method, is added to response comments by Reviewer 2 and 3.
5. Appendix B discusses results from the exact sampler is added to response a comment by Reviewer 3.
6. Appendix I is added to supplement a newly added Section 5.
7. We have rearranged Appendices and resized some figures for better readability.

---

## Editorial Decision

editor-in-charge_assigned